# Log-Concave and Multivariate Canonical Noise Distributions for Differential Privacy

**Jordan A. Awan**
Department of Statistics
Purdue University
jawan@purdue.edu

**Jinshuo Dong**
Department of Computer Science
Northwestern University and IDEAL[*]
jinshuo@northwestern.edu

## Abstract

A canonical noise distribution (CND) is an additive mechanism designed to satisfy $f$-differential privacy ($f$-DP), without any wasted privacy budget. $f$-DP is a hypothesis testing-based formulation of privacy phrased in terms of *tradeoff functions*, which captures the difficulty of a hypothesis test. In this paper, we consider the existence and construction of both log-concave CNDs and multivariate CNDs. Log-concave distributions are important to ensure that higher outputs of the mechanism correspond to higher input values, whereas multivariate noise distributions are important to ensure that a joint release of multiple outputs has a tight privacy characterization. We show that the existence and construction of CNDs for both types of problems is related to whether the tradeoff function can be decomposed by functional composition (related to group privacy) or mechanism composition. In particular, we show that pure $\epsilon$-DP cannot be decomposed in either way and that there is neither a log-concave CND nor any multivariate CND for $\epsilon$-DP. On the other hand, we show that Gaussian-DP, $(0, \delta)$-DP, and Laplace-DP each have both log-concave and multivariate CNDs.

## 1 Introduction

Differential privacy (DP), proposed by Dwork et al. [2006], is the state-of-the-art framework in formal privacy protection and is being implemented by tech companies, government agencies, and academic institutions. Over time, the DP community has developed many new DP mechanisms as well as new frameworks. Recently, $f$-DP (Dong et al. [2022] was proposed as a generalization of DP, allowing for tight calculations of group privacy, composition, subsampling, and post-processing. It was shown in Dong et al. [2022] that $f$-DP is provably the tightest version of DP that respects the post-processing property of DP. In particular, $f$-DP can be losslessly converted to Rényi-DP (or any $f$-divergence version of DP) as well as $(\epsilon, \delta)$-DP, but not vice-versa [Dong et al., 2022]. Furthermore, $f$-DP is equivalent (can be losslessly converted back and forth) to the privacy profile [Balle et al., 2018, 2020] and the privacy loss random variables [Sommer et al., 2019, Zhu et al., 2022].

$f$-DP is defined in terms of a *tradeoff function* or *receiver operator curve (ROC)*, which encapsulates the difficulty of conducting a hypothesis test between two distributions. If $f = T(P, Q)$ is the tradeoff function for testing between the distributions $P$ and $Q$, then if a mechanism $M$ satisfies $f$-DP, this means that given the output of $M$ when run on one of two adjacent databases, it is at least as hard to determine which database was used, as it is to test between $P$ and $Q$.

While $f$-DP has the many desirable theoretical properties listed above in its favor, there are limited techniques for working with $f$-DP, and few constructive mechanisms for an arbitrary $f$-DP guarantee. A notable exception is a *canonical noise distribution* (CND) from the recent paper Awan and Vadhan

---

[*]The Institute for Data, Econometrics, Algorithms, and Learning

[2021], which builds a one-dimensional additive noise mechanism designed to exactly satisfy $f$-DP, with no wasted privacy budget. Along with the intuitive idea that a CND is optimal in that it optimizes the privacy loss budget, Awan and Vadhan [2021] showed that CNDs are crucial to the construction of optimal DP hypothesis tests and free DP $p$-values. However, the CND construction given in Awan and Vadhan [2021] does not result in a smooth distribution, and in particular is not *log-concave*. Log-concavity is a desirable property because it implies that the distribution has a monotone likelihood ratio; this means that higher observed values are always more likely to have come from a higher input value than a lower one. Log-concavity thus makes the DP output much more interpretable, easily analyzed, and also has makes the calculation of the privacy cost simpler [Dong et al., 2021]. Furthermore, the results of Awan and Vadhan [2021] are limited to 1-dimensional distributions.

In this paper, we develop new properties of CNDs and $f$-DP, motivated by the following two questions,

1. Can we construct log-concave CNDs?   2. Can we construct multivariate CNDs?

**Our Contributions** The existence of both log-concave 1-dimensional CNDs and multivariate CNDs are intricately linked with properties related to group privacy and mechanism composition. Two highly desirable properties of a tradeoff function are *infinite divisibility* and *infinite decomposability*, meaning that the tradeoff function can be exactly achieved by $n$-fold group privacy or $n$-fold mechanism composition, respectively. We prove that a tradeoff function has a log-concave CND if and only if the tradeoff function is infinitely divisible, and give a construction for the *unique* log-concave CND in this case. We also show that if a tradeoff function is either infinitely divisible or decomposable, then we can construct a multivariate CND.

Along with the positive results listed above, we also include impossibility results. In particular, $(\epsilon, 0)$-DP is neither divisible nor decomposable, and in fact has neither a log-concave CND nor any multivariate CND. In contrast to $(\epsilon, 0)$-DP, two families that satisfy both infinite divisibility and infinite decomposability are $\mu$-GDP and $(0, \delta)$-DP. While $(0, \delta)$-DP has limited applicability due to its weak protection for events with small probability, $\mu$-GDP and related DP definitions (such as zero concentrated DP) have been gaining popularity. The results of this paper provide a new perspective supporting the adoption of GDP as the default privacy measure instead of $(\epsilon, 0)$-DP.

**Organization** In Section 2, we review concepts in $f$-DP and canonical noise distributions. In Section 3, we study 1-dimensional CNDs. In Section 3.1, we prove that the Tulap distribution is the *unique* CND for $(\epsilon, 0)$-DP. In Section 3.2, we propose the concept of infinite divisibility and prove that a tradeoff function has a log-concave CND if and only if it is infinitely divisible; we also give a construction to produce the log-concave CND from a family of infinitely divisible tradeoff functions. We prove that piece-wise linear tradeoff functions are generally not infinitely divisible in Section 3.3, and in particular $(\epsilon, 0)$-DP and several related tradeoff functions do not have log-concave CNDs. In Section 4, we propose a multivariate extension of CND. We give two general constructions of multivariate CNDs in Section 4.1 depending on whether a tradeoff function is decomposable or infinitely divisible. We give several examples of multivariate CNDs in Sections 4.2-4.5 for Gaussian DP, $(0, \delta)$-DP, $(\epsilon, \delta)$-DP, and Laplace-DP. In Section 4.6, we show that there is no multivariate CND for $(\epsilon, 0)$-DP, which implies that $(\epsilon, 0)$-DP is not decomposable. We conclude with discussion in Section 5. Proofs and technical details are found in the Appendix.

**Related Work** While there are many complex DP mechanisms, many use the fundamental building block of additive mechanisms (e.g., functional mechanism [Zhang et al., 2012], objective perturbation [Chaudhuri et al., 2011, Kifer et al., 2012], stochastic gradient descent [Abadi et al., 2016], and the sparse vector technique [Dwork et al., 2009, Zhu and Wang, 2020], to name a few). There have been many different additive mechanisms proposed in the literature, for different privacy purposes. We highlight the works that show some optimality property for the proposed noise distributions. This work is most directly building off of Awan and Vadhan [2021], who proposed the concept of canonical noise distributions as a method of quantifying what it means to fully use the privacy budget. There are also other works, which derive optimal mechanisms with respect to other metrics. Ghosh et al. [2012] showed that a discrete Laplace distribution is the universal utility maximizer for a general class of utility functions in pure-DP. Geng and Viswanath [2015b] proposed the staircase mechanism which they showed optimizes the $\ell_1$ or $\ell_2$ error for pure-DP. For $(\epsilon, \delta)$-DP, Geng and Viswanath [2015a] showed that either the staircase or a uniform distribution can achieve the optimal rate in terms of $\ell_1$ and $\ell_2$ error. Steinke and Ullman [2016] showed that the $\ell_\infty$-mechanisms is rate optimal when measuring utility in terms of $\ell_\infty$ error. Awan and Slavković [2020] derive optimal mechanisms

among the class of $K$-Norm Mechanisms, proposed by Hardt and Talwar [2010], in terms of various scale-independent measures, for a fixed statistic and sample size.

## 2  Differential privacy basics

Differential privacy ensures that given the output of a private mechanism, it is difficult for an adversary to determine whether an individual is present in the database or not. To satisfy DP, a privacy expert employs a *mechanism $M$*, which is a set of probability distributions $M_D$ on a common space $\mathscr{Y}$, indexed by possible databases $D \in \mathscr{D}$. Let $d(D, D')$ be an integer-valued metric on the space of databases $\mathscr{D}$, which represents the number of entries that $D$ and $D'$ differ in. We call $D$ and $D'$ *adjacent* if $d(D, D') \leq 1$. While there are now many variants of DP, they all center around the idea that given a randomized algorithm $M$, for any two adjacent databases $D$, $D'$, the distributions of $M(D)$ and $M(D')$ should be "similar." While many DP variants measure similarity in terms of divergences, $f$-DP formalizes similarity in terms of hypothesis tests. Intuitively, for two adjacent databases $D$ and $D'$, a mechanism $M$ satisfies $f$-DP if given the output of $M$, it is difficult to determine whether the original database was $D$ or $D'$. This is formalized in terms *tradeoff functions*.

For two distributions $P$ and $Q$, the *tradeoff function* (or ROC) between $P$ and $Q$ is $T(P, Q) : [0, 1] \to [0, 1]$, where $T(P, Q)(\alpha) = \inf\{1 - \mathbb{E}_Q \phi \mid \mathbb{E}_P(\phi) \geq 1 - \alpha\}$, where the infinimum is over all measurable tests $\phi$. The tradeoff function returns the optimal type II error for testing $H_0 = P$ versus $H_1 = Q$ at specificity (one minus type I error) $\alpha$, and captures the difficulty of distinguishing between $P$ and $Q$. [2] A function $f : [0, 1] \to [0, 1]$ is a tradeoff function if and only if $f$ is convex, continuous, non-decreasing, and $f(x) \leq x$ for all $x \in [0, 1]$ [Dong et al., 2022, Proposition 2.2]. We say that a tradeoff function $f$ is *nontrivial* if $f(\alpha) < \alpha$ for some $\alpha \in (0, 1)$.

**Definition 2.1** ($f$-DP: Dong et al., 2022). Let $f$ be a tradeoff function. A mechanism $M$ satisfies $f$-DP if $T(M(D), M(D')) \geq f$, for all $D, D' \in \mathscr{D}$ which satisfy $d(D, D') \leq 1$.

Intuitively, a mechanism satisfies $f$-DP, where $f = T(P, Q)$, if testing $H_0 : M(D)$ versus $H_1 : M(D')$ is at least as hard as testing $H_0 : P$ versus $H_1 : Q$. Without loss of generality we can assume that $f$ is *symmetric*, meaning that if $f = T(P, Q)$, then $f = T(Q, P)$. This is due to the fact that adjacency of databases is a symmetric relation [Dong et al., 2022, Proposition 2.4]. So, we limit the focus of this paper on symmetric tradeoff functions.

A key property of differential privacy is that it also implies privacy guarantees for groups. Dong et al. [2022, Theorem 2.14] showed that if a mechanism is $f$-DP, then it satisfies $f^{\circ k}$-DP, when the adjacency measure is changed to allow for a difference in $k$ entries (where $f^{\circ k}$ means the functional composition of $f$ with itself, $k$ times). We call this *group privacy*, which is a central topic in differential privacy. Note that the bound $f^{\circ k}$ is not necessarily the tightest privacy guarantee for a particular mechanism.

*Mechanism Composition* quantifies the cumulative privacy cost of the output of $k$ mechanisms. To express the tradeoff function resulting from composition, Dong et al. [2022] proposed the *tensor product* of tradeoff functions: if $f = T(P, Q)$ and $g = (P', Q')$, then $f \otimes g := T(P \times P', Q \times Q')$, which they show is well defined, commutative, and associative. They prove that if we have $k$ mechanisms $M_1, \ldots, M_k$, which each satisfy $f_1$-DP, $f_2$-DP,..., $f_k$-DP respectively, then the composition $(M_1, \ldots, M_k)$ satisfies $f_1 \otimes \cdots \otimes f_k$-DP (see Dong et al. [2022, Theorem 3.2] for a more precise statement).

The traditional framework of $(\epsilon, \delta)$-DP is a subclass of $f$-DP: Let $\epsilon \geq 0$ and $\delta \in [0, 1]$. A mechanism satisfies $(\epsilon, \delta)$-DP if it satisfies $f_{\epsilon, \delta}$-DP, where $f_{\epsilon, \delta}(\alpha) = \max\{0, 1 - \delta - e^\epsilon + e^\epsilon \alpha, \exp(-\epsilon)(\alpha - \delta)\}$. An important special case is $(\epsilon, 0)$-DP, which was the original definition of DP.

Another popular subclass is Gaussian-DP (GDP): For $\mu \geq 0$, a mechanism satisfies $\mu$-GDP if it satisfies $G_\mu$-DP, where $G_\mu = T(N(0, 1), N(\mu, 1))$. Gaussian-DP was proposed in Dong et al. [2022] and has several desirable properties, such as being closed under group privacy and closed under composition. Dong et al. [2022] also established a central limit theorem for tradeoff functions as the number of compositions approaches infinity, showing that under general assumptions the tradeoff function of the composed mechanisms approaches $G_\mu$ for some $\mu$.

---

[2]In Dong et al. [2022], the tradeoff function was originally defined as a function of type I error. Our choice to flip the tradeoff function along the $x$-axis is for mathematical convenience. The ROC function is usually defined as the power (one minus type II error) as a function of type I error.

## 2.1 Canonical noise distributions

To satisfy DP, additive mechanisms must introduce noise proportional to the *sensitivity* of the statistic of interest. Let $\|\cdot\|$ be a norm on $\mathbb{R}^d$. A statistic $S : \mathscr{D} \to \mathbb{R}^d$ has $\|\cdot\|$-*sensitivity* $\Delta > 0$ if $\|S(D) - S(D')\| \leq \Delta$ for all $d(D, D') \leq 1$. When $d = 1$, we use $|\cdot|$ as the default norm. Any additive mechanism, which releases $S(D) + \Delta N$, satisfies $f$-DP if $T(N, N + v) \geq f$ for all $\|v\| \leq 1$. The concept *canonical noise distribution* (CND) was proposed by Awan and Vadhan [2021] to capture when an additive mechanism satisfies $f$-DP, and "fully uses the privacy budget."

**Definition 2.2** (Canonical noise distribution: Awan and Vadhan [2021])**.** Let $f$ be a symmetric tradeoff function. A continuous random variable $N$ with cumulative distribution function (cdf) $F$ is a *canonical noise distribution* (CND) for $f$ if

1. For any $m \in [0, 1]$, $T(N, N + m) \geq f$,

2. $f(\alpha) = T(N, N + 1)(\alpha)$ for all $\alpha \in (0, 1)$,

3. $T(N, N + 1)(\alpha) = F(F^{-1}(\alpha) - 1)$ for all $\alpha \in (0, 1)$,

4. $F(x) = 1 - F(-x)$ for all $x \in \mathbb{R}$; that is, $N$ is symmetric about zero.

In Definition 2.2, property 1 ensures that the additive mechanism using a CND satisfies $f$-DP, property 2 ensures that the privacy guarantee is tight, property 3 gives a closed form for the tradeoff function in terms of the CND's cdf, which is equivalent to enforcing a monotone likelihood ratio property, and property 4 imposes symmetry which is mostly for convenience.

An important property of CNDs is that they satisfy the following recurrence relation:

**Lemma 2.3** (Awan and Vadhan [2021])**.** *Let $f$ be a symmetric nontrivial tradeoff function and let $F$ be a CND for $f$. Then $F(x) = 1 - f(1 - F(x - 1))$ when $F(x - 1) > 0$ and $F(x) = f(F(x + 1))$ when $F(x + 1) < 1$.*

In Awan and Vadhan [2021], they showed that the above recurrence relation can be used to construct a CND for any nontrivial symmetric tradeoff function.

**Proposition 2.4** (CND construction: Awan and Vadhan [2021])**.** *Let $f$ be a symmetric nontrivial tradeoff function, and let $c \in [0, 1]$ be the solution to $f(1 - c) = c$. We define $F_f : \mathbb{R} \to \mathbb{R}$ as*

$$F_f(x) = \begin{cases} f(F_f(x + 1)) & x < -1/2 \\ c(1/2 - x) + (1 - c)(x + 1/2) & -1/2 \leq x \leq 1/2 \\ 1 - f(1 - F_f(x - 1)) & x > 1/2. \end{cases}$$

*Then $N \sim F_f$ is a canonical noise distribution for $f$.*

While Proposition 2.4 gives a general construction of a CND for an arbitrary $f$, the resulting distribution is generally not smooth or log-concave. Awan and Vadhan [2021] showed that in the case of $G_\mu$, this construction does not recover the Gaussian distribution, which is the log-concave CND.

# 3 One-dimensional CNDs

In this section, we expand on the results of Awan and Vadhan [2021], by producing new results for one-dimensional CNDs. In Section 3.1, we show that the Tulap distribution is the *unique* CND for $(\epsilon, 0)$-DP. In Section 3.2, we propose the concept of an *infinitely divisible tradeoff function* and show that a tradeoff function has a log-concave CND if and only if it is infinitely divisible. We also give a construction to produce the unique log-concave CND for an infinitely divisible family of tradeoff functions. In Section 3.3, we determine when a piece-wise linear tradeoff function is divisible, and show that $f_{\epsilon,0}$ and related tradeoff functions are not infinitely divisible, and hence do not have log-concave CNDs.

## 3.1 CNDs for $(\epsilon, 0)$-DP

In Awan and Vadhan [2021], it was shown that in general, the CND is not unique, but it was not clear whether there existed alternative CNDs for $f_{\epsilon,0}$ or $f_{\epsilon,\delta}$. We begin this section by showing that

the Tulap distribution, which was shown to be a CND for $f_{\epsilon,\delta}$ by Awan and Vadhan [2021] is in fact the *unique* CND for $f_{\epsilon,0}$. The Tulap distribution was proposed by Awan and Slavković [2018] for the purpose of designing uniformly most powerful hypothesis tests for Bernoulli data. In the case of $(\epsilon,0)$-DP, the Tulap distribution coincides with one of the staircase mechanisms [Geng and Viswanath, 2015b]. It is also closely related to the discrete Laplace distribution (also known as the geometric mechanism), which is optimal for a wide range of utility functions in Ghosh et al. [2012].

**Proposition 3.1.** *Let $\epsilon > 0$. The distribution* $\mathrm{Tulap}(0, \exp(-\epsilon), 0)$ *is the unique CND for $f_{\epsilon,0}$.*

*Proof Sketch.* By Lemma 2.3, the only choice in a CND is on $[-1/2, 1/2]$. If the density is non-constant on $[-1/2, 1/2]$, we show that the likelihood ratio is not bounded by $e^\epsilon$, violating $\epsilon$-DP. $\square$

Proposition 3.1 is a surprising result in that one may expect a more natural CND than the Tulap distribution, which has a discontinuous density. However, we now know that there are no other CNDs for $(\epsilon, 0)$-DP. In particular, there is no log-concave CND, which is the topic of the next subsection.

## 3.2 Infinite divisibility and log-concavity

It has been shown in Dong et al. [2022] and Dong et al. [2021] that tradeoff functions built from location family log-concave distributions have very nice properties for $f$-DP. Log-concave distributions are have a monotone likelihood ratio property which gives a simple closed form expression for the tradeoff function in terms of the cdf of the log-concave distribution. It is easily observed that a tradeoff function with a log-concave CND satsifies a property that we call *infinite divisibility*. We prove that in fact a tradeoff function has a log-concave CND if and only if it is infinitely divisble. Our proof also results in a construction to produce the unique log-concave CND.

A continuous random variable $X$ is *log-concave* if its density can be written as $g_X(x) \propto \exp(C(x))$, where $C$ is a concave function. We call a (symmetric) tradeoff function $f$ *log-concave* if there exists a log-concave CND $N$ for $f$. Recall that if $N \sim F$ is a CND for $f$, then $f(\alpha) = F(F^{-1}(\alpha) - 1)$. If $N$ is also log-concave, then $f_t(\alpha) := F(F^{-1}(\alpha) - t)$ is a tradeoff function for every $t \in [0, \infty)$, and the family $\{f_t \mid t \in [0, \infty)\}$ is a monoid satisfying the assumptions of Definition 3.2.

**Definition 3.2.** A tradeoff function $f$ is *infinitely divisible* if there exists a monoid, under the operation of functional composition, $\{f_t \in \mathscr{F} \mid t \geq 0\}$ containing $f$ such that

1. $f_t \circ f_s = f_{t+s}$ for all $s, t \geq 0$,

2. $f_s$ is nontrivial for all $s > 0$, and

3. $f_s \to f_0 = \mathrm{Id}$ as $s \downarrow 0$.

The discussion above established that log-concave CNDs are infinitely divisible. The key result of this section is that a tradeoff function is log-concave if and only if it is infinitely divisible. We saw that it is easy to construct the infinitely divisible family given a log-concave CND. Surprisingly, we give a construction to derive the log-concave CND from the infinitely divisible family as well. This result shows an intimate relationship between properties of a tradeoff function and the possible CNDs for that tradeoff function. We will see in Section 4.1 that the property of infinite divisibility shows up again in the construction of multivariate CNDs.

**Theorem 3.3.** *A nontrivial tradeoff function $f \in \mathscr{F}$ is log-concave if and only if it is infinitely divisible. In particular,*

1. *If $f$ is log-concave with log-concave CND $N \sim F$, then $\{f_t \mid t \geq 0\}$ defined by $f_t = F(F^{-1}(\alpha) - t)$ satisfies the assumptions of Definition 3.2.*

2. *Let $f$ be infinitely divisible, with monoid $\{f_t \in \mathscr{F} \mid t \geq 0\}$, as defined in Definition 3.2, such that $f = f_1$. Let $F_s$ be any CND for $f_s$ (such as constructed in Proposition 2.4). Then the following limit exists $F^*(t) := \lim_{s \to 0} F_s(\frac{1}{s}t)$ and $N \sim F^*$ is the unique log-concave CND for $f$. Furthermore, $F^*(st)$ is the unique log-concave CND for $f_s$, for all $s > 0$.*

*Proof Sketch.* It is easy to verify property 1. For property 2, we consider a subsequence $s_n = 1/n!$ and observe that $F_{1/n!}(n!t)$ is a CND for $f$ at every $n$, but that as $n$ increases, the number of points

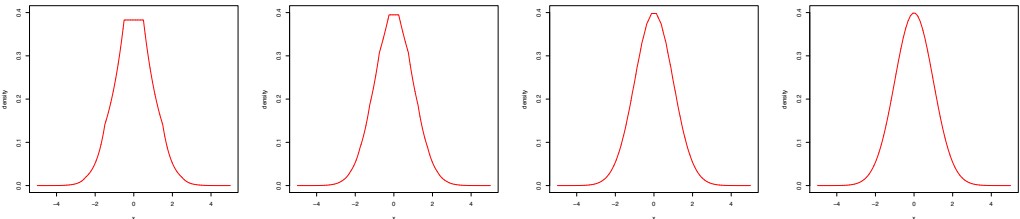

Figure 1: An illustration of Theorem 3.3 when applied to $G_1 = T(N(0,1), N(1,1))$. From left to right, we have the density corresponding to $F_{G_{2^{-n}}}(2^n t)$ for $n = 0, 1, 2, 3$.

at which the CND is uniquely determined also increases, by Lemma 2.3. In the limit, this sequence converges to a unique cdf, which we show has the properties of a log-concave CND. □

**Example 3.4.** We will illustrate the limit of Theorem 3.3 on $G_1$. Let $F_{G_{2^{-n}}}$ be the constructed cdf from Proposition 2.4 for $n = 0, 1, 2, 3$. The density functions corresponding to $F_{G_{2^{-n}}}(2^n t)$ are plotted in Figure 1. We see that as $n$ increases, the pdfs approach that of a standard normal, which we know is the log-concave CND for $f = G_1$.

When the construction of Theorem 3.3 is applied to $f_{\epsilon,0}$, the cdf $F^*$ converges to a Laplace cdf. This seems to reflect the fact that under the limit of group privacy, $(\epsilon, 0)$-DP converges to Laplace-DP Dong et al. [2022, Proposition 2.15].

Finally, we illustrate why properties 2 and 3 of Definition 3.2 are necessary for Theorem 3.3.

**Example 3.5** (Non examples for Theorem 3.3). First consider why it is necessary to have $f_s \to \mathrm{Id}$. Set $f_s(\alpha) = I(\alpha = 1)$ for all $s > 0$. Note that $f_s \circ f_t = f_{s+t}$, but that the construction of Theorem 3.3 results in a point mass at zero, which is not a CND as it is not continuous.

Next, suppose that all of the tradeoff functions are trivial, then $f_s(\alpha) = \alpha$ for all $s > 0$, and $f_s \circ f_t = f_{s+t}$. However, there are no CNDs in this case.

### 3.3 Piece-wise linear tradeoff functions are generally not infinitely divisible

We showed in Theorem 3.3 that if a tradeoff function is infinitely divisible, then we can construct a log-concave CND. However, it is not always obvious whether a tradeoff function is infinitely divisible or not. We show that in the case of piece-wise linear tradeoff functions, we can upper bound the number of possible divisions in terms of the number of break points. In particular, the piece-wise linear tradeoff functions considered in this section are not infinitely divisible.

We can characterize the piece-wise linear convex functions in terms of the 2nd derivative behavior: A convex function is piece-wise linear if and only if its 2nd derivative is defined everywhere except for finitely many points, and is zero whenever it is defined.

Part 1 of Proposition 3.6 shows that a piece-wise linear tradeoff function $f$, which satisfies $f(x) = 0$ implies $x = 0$, can be sub-divided only a finite number of times. A consequence of this is that $f_{\epsilon,0}$ and several related tradeoff functions are not infinitely divisible and hence do not have log-concave CNDs. In fact, not only is $f_{\epsilon,0}$ not infinitely divisible, but there is in fact *no division* $f_{\epsilon,0} = f \circ g$ into symmetric tradeoff functions, except where either $f$ or $g$ is the identity!

**Proposition 3.6.**    *1. Let $f$ be a nontrivial piece-wise linear tradeoff function with $k \geq 1$ breakpoints and such that $f(x) = 0$ implies that $x = 0$. Then there is no tradeoff function $g$ such that $g^{\circ(k+1)} = f$.*

   *2. Let $\epsilon > 0$. There does not exist nontrivial symmetric tradeoff functions $f_1$ and $f_2$ such that $f_{\epsilon,0} = f_1 \circ f_2$.*

   *3. Let $f$ be the tradeoff function obtained by an arbitrary sequence of mechanism compositions, functional compositions, or subsampling (without replacement) of $f_{\epsilon,0}$ (could be different $\epsilon$ values for each). Then $f$ is not infinitely divisible and so does not have a log-concave CND.*

*Proof Sketch.* We show in Lemma A.12 that divisions of a piece-wise linear tradeoff function are themselves piece-wise linear, and that the functional composition of piece-wise linear tradeoff functions increases the number of breakpoints. This then limits the number of divisions a piece-wise linear tradeoff function can have in terms of the number of its breakpoints. □

**Example 3.7** ($f_{0,\delta}$ is log-concave). What if $f(x) = 0$ does not imply that $x = 0$? The tradeoff functions $f_{0,\delta}$ fit within this setting, and the results of Proposition 3.6 do *not* apply here. In fact, $f_{0,\delta}$ is infinitely divisible with log-concave CND $U(-1/(2\delta), 1/(2\delta))$. That is $f_{0,\delta} = T(U, U + \delta)$ where $U \sim U(-1/2, 1/2)$. While $f_{\epsilon,\delta}$ for $\delta > 0$ also does not satisfy the assumption that $f_{\epsilon,\delta}(x)$ implies $x = 0$, it is not clear at this time whether $f_{\epsilon,\delta}$ is log-concave or not.

# 4 Multivariate CNDs

In this section, we generalize the definition of CND to dimensions greater than one. While in the univariate case, *sensitivity* is measured using the absolute distance between two statistic values, in $\mathbb{R}^d$, there are many choices of norms which can be used to measure the sensitivity [Awan and Slavković, 2020]. So, we will specify the sensitivity norm when talking about a multivariate CND. In Definition 4.1 we define a multivariate CND to be a natural generalization of properties 1-4 of Definition 2.2.

**Definition 4.1.** Let $f$ be a symmetric tradeoff function, and let $\|\cdot\|$ be a norm on $\mathbb{R}^d$. A continuous random vector $N$ with density $g$ is a *canonical noise distribution* (CND) for $f$, with respect to $\|\cdot\|$, if

1. For all $v \in \mathbb{R}^d$ such that $\|v\| \leq 1$ we have that $T(N, N + v) \geq f$,

2. there exists $\|v^*\| \leq 1$ such that $T(N, N + v^*)(\alpha) = f(\alpha)$ for all $\alpha \in (0, 1)$,

3. for all $v^*$ which satisfy property 2, and all $w \in \mathbb{R}^d$, we have that the likelihood ratio $g(w + tv^* - v^*)/g(w + tv^*)$ is a non-decreasing function of $t \in \mathbb{R}$,

4. $N$ is symmetric about zero: $g(x) = g(-x)$ for all $x \in \mathbb{R}^d$.

When restricted to $d = 1$, Definition 4.1 recovers Definition 2.2. This is clear for properties 1, 2, and 4. Property 3 of Definition 2.2 can be interpreted as requiring that an optimal rejection set for $T(N, N + 1)$ is of the $[x, \infty)$ for some $x$. By the Neyman Pearson Lemma, we know that this holds if and only if the likelihood ratio $F'(x - 1)/F'(x)$ is non-decreasing in $x$. We see that when $d = 1$, property 3 of Definition 4.1 is equivalent to property 3 of Definition 2.2. We can interpret Property 3 of Definition 4.1 as enforcing a monotone likelihood ratio in directions parallel to $v^*$.

## 4.1 Constructions of multivariate CNDs

Composition gives a simple method to construct a multivariate CND whenever a tradeoff function can be decomposed into the composition of $k$ tradeoff functions:

**Proposition 4.2.** *Suppose that $f = f_1 \otimes f_2 \otimes \cdots \otimes f_k$ all be nontrivial and symmetric tradeoff functions, and let $F_1, F_2, \ldots, F_k$ be CNDs for $f_1, \ldots, f_k$ respectively. Let $N = (N_1, \ldots, N_k)$ be the random vector where $N_i \sim F_i$ are independent. Then $N$ is a CND for $f$ with respect to $\|\cdot\|_\infty$.*

Interestingly, when a tradeoff function is infinitely divisible and hence has a log-concave CND by Theorem A.8, we can create a multivariate CND with respect to $\|\cdot\|_1$-sensitivity.

**Theorem 4.3.** *Let $f$ be a nontrivial and symmetric log-concave tradeoff function with log-concave CND $F$. Let $N = (N_1, \ldots, N_k)$ be the random vector where $N_i \sim F$ are independent. Then $N$ is a (log-concave) CND for $f$ with respect to $\|\cdot\|_1$.*

*Proof Sketch.* Since the noise added is i.i.d., we can rephrase the tradeoff function as the tensor product of the individual tradeoff functions. We apply Theorem A.2 which lower bounds the tensor product of tradeoff functions with the functional composition. □

Theorem 4.3 was inspired by the i.i.d. Laplace mechanism. In Section 4.5, we show that the i.i.d. Laplace mechanism is a special case of Theorem 4.3 and gives a multivariate CND for Laplace-DP.

Note that Theorem 4.3 results in a log-concave multivariate CND, and if each of $N_1, \ldots, N_k$ are log-concave in Proposition 4.2, then that constructed multivariate CND is log-concave as well. [Dong

et al. [2021] showed that log-concave distributions have many nice properties in multivariate settings as well. We leave it to future work to investigate when multivariate log-concave CNDs exist.

## 4.2 Multivariate CND for GDP

Recall that if $N \sim N(0, I)$ is a $d$-dimensional Gaussian random vector, and $v \in \mathbb{R}^d$ is any vector, then $T(N, N + v) = T(N(0, 1), N(\|v\|_2, 1))$ [Dong et al., 2022, Proposition D.1(5)]. This previous result implies that $N(0, I)$ was a multivariate CND for GDP under $\|\cdot\|_2$-sensitivity. In fact, we show in Proposition 4.4 that for GDP, any multivariate Gaussian is a CND with respect to any norm.

**Proposition 4.4.** *Let $\Sigma$ be a $d \times d$ positive definite matrix. Let $v^* \in \text{argmax}_{\|u\| \leq 1} \|\Sigma^{-1/2}u\|_2$. Then $N(0, \Sigma)$ is a $d$-dimensional CND for $\|\Sigma^{-1/2}v^*\|_2$-GDP with respect to the norm $\|\cdot\|$.*

**Remark 4.5.** While a multivariate Gaussian is always a multivariate CND for GDP, there is still possibly room for improvement. For Definition 4.1, we only need a single vector to satisfy property 2. However, we could potentially ask that the bound is achieved at all $u$ such that $\|u\| = 1$. Note that if $\|\cdot\|$ is an elliptical norm, then we do get this stronger property for the multivariate Gaussian, when we choose $\Sigma$ to align with the sensitivity norm.

## 4.3 Multivariate CND for $(0, \delta)$-DP

First let's review a few facts about $(0, \delta)$-DP, also known as $f_{0,\delta}$-DP. First, note that $U(\frac{-1}{2\delta}, \frac{1}{2\delta})$ is a (log-concave) CND for $f_{0,\delta}$. So, we can write $f_{0,\delta} = T(U, U + \delta)$ where $U \sim U(-1/2, 1/2)$. Because of this, we have that $f_{0,\delta}$ is infinitely divisible, and $f_{0,\delta_1} \circ f_{0,\delta_2} = f_{0,\min\{\delta_1+\delta_2,1\}}$. Furthermore, $f_{0,\delta_1} \otimes f_{0,\delta_2} = f_{0,1-(1-\delta_1)(1-\delta_2)}$, as observed in Dong et al. [2022]. This means that $f_{0,\delta}$ is also infinitely decomposable, a property that we had only seen for GDP before. This decomposability implies, by Proposition 4.2 that we can build a multivariate CND for $f_{0,\delta}$ under $\|\cdot\|_\infty$-sensitivity. In fact, this construction is a multivariate CND for any sensitivity norm.

**Proposition 4.6.** *Let $0 < \delta \leq 1$, $d \geq 1$, and $\|\cdot\|$ be a norm on $\mathbb{R}^d$. Call $v^* \in \arg \min_{\|v\| \leq 1} \prod_{i=1}^{d}(1 - \delta|v_i|)$ and $A = \prod_{i=1}^{d}(1 - \delta|v_i^*|)$. Then $U(\frac{-1}{2\delta}, \frac{1}{2\delta})^n$ is a CND for $f_{0,1-A}$ under $\|\cdot\|$-sensitivity. In the special case of $\|\cdot\| = \|\cdot\|_\infty$, this simplifies to $A = (1 - \delta)^d$.*

## 4.4 Multivariate CND for $f_{\epsilon,\delta}$ when $\delta > 0$

Let $\epsilon > 0$ and $\delta \in (0, 1]$. Recall that $f_{\epsilon,\delta} = f_{\epsilon,0} \otimes f_{0,\delta}$ [Dong et al., 2022]. Since $f_{0,\delta}$ is infinitely decomposable, we can write $f_{\epsilon,\delta} = f_{\epsilon,0} \otimes f_{0,\delta_1} \otimes \cdots \otimes f_{0,\delta_k}$ where $\delta = \prod_{i=1}^{k}(1 - \delta_i)$. By Proposition 4.2 we construct a multivariate CND for $f_{\epsilon,\delta}$ with respect to $\|\cdot\|_\infty$-sensitivity by using $\text{Tulap}(0, \exp(-\epsilon), 0)$ in one coordinate, and the uniform distributions $U(\frac{-1}{2\delta_i}, \frac{1}{2\delta_i})$ in the other $k$ coordinates.

## 4.5 Two multivariate CNDs for Laplace-DP

Many mechanisms designed to satisfy $(\epsilon, 0)$-DP actually satisfy the stronger privacy guarantee of Laplace-DP. In particular, variations on the Laplace mechanism are very common additive mechanisms used to achieve $(\epsilon, 0)$-DP. In this section, we show that two multivariate versions of the Laplace mechanism, the $\ell_1$ and $\ell_\infty$ mechanisms, are multivariate CNDs for Laplace-DP.

The Laplace distribution, denoted $\text{Laplace}(m, s)$ is a distribution on $\mathbb{R}$ with density $\frac{1}{2s} \exp(\frac{-1}{s}|x - m|)$. We say a mechanism satisfies $\epsilon$-Laplace-DP if it satisfies $L_\epsilon$-DP, where $L_\epsilon := T(N, N + \epsilon)$ and $N \sim \text{Laplace}(0, 1)$. It is easily seen that $\text{Laplace}(0, 1/\epsilon)$ is a log-concave CND for $\epsilon$-Laplace-DP.

**i.i.d. Laplace Mechanism** The i.i.d. Laplace mechanism is defined as follows: Let $\epsilon > 0$ be given. If $T : \mathcal{X} \to \mathbb{R}^k$ has $\|\cdot\|_1$-sensitivity of $\Delta$, then the i.i.d. Laplace mechanism releases $T(X) + \Delta N$, where $N = (N_1, \ldots, N_k)$ is the random vector with i.i.d. entries $N_i \sim \text{Laplace}(0, 1/\epsilon)$. It is well known that the i.i.d. Laplace mechanism satisfies $f_{\epsilon,0}$-DP [Dwork et al., 2014, Theorem 3.6]. Since $N_1$ is a log-concave CND for $L_\epsilon$, Theorem 4.3 shows that $N$ is a CND for $L_\epsilon$, with respect to $\|\cdot\|_1$-sensitivity. As $L_\epsilon \geq f_{\epsilon,0}$ and $L_\epsilon(\alpha) > f_{\epsilon,0}(\alpha)$ for some values of $\alpha$, we can more precisely capture the privacy cost of the i.i.d. Laplace mechanism using tradeoff functions rather than $\epsilon$-DP.

$\ell_\infty$-**Mechanism** The $\ell_\infty$-mechanism, proposed in Steinke and Ullman [2016] is a special case of the $K$-norm mechanisms [Hardt and Talwar, 2010], with density proportional to $\exp(-\epsilon\|x\|_\infty)$. Steinke and Ullman [2016] showed that the $\ell_\infty$ mechanism can improve the sample complexity of answering multiple queries, when accuracy is measured by $\ell_\infty$-norm. Awan and Slavković [2020] showed that the $\ell_\infty$ mechanism is near optimal in certain applications of private linear and logistic regression. It is well known that when using $\ell_\infty$-sensitivity, the $\ell_\infty$-mechanism satisfies $\epsilon$-DP. In this section, we show that the $\ell_\infty$-mech is a CND for $L_\epsilon$, with respect to $\ell_\infty$-sensitivity.

**Proposition 4.7.** *Let $\epsilon > 0$, and $d \geq 1$. Let $X$ be a $d$-dimensional random vector with density $g(x) = \frac{\exp(-\epsilon\|x\|_\infty)}{d!(2/\epsilon)^d}$. Then $X$ is a CND for the tradeoff function $L_\epsilon$ with respect to $\|\cdot\|_\infty$.*

*Proof Sketch.* First we show that with the shift of $v^* = (1, 1, \ldots, 1)$, the privacy loss random variable coincides with that of $L_\epsilon$. Then, we show that $v^* = (1, 1, \ldots, 1)^\top$ is the worst case of any shift $v$ to minimize the tradeoff functions. To deal with the case that some of the entries of $v$ are zero, we establish a convergence theorem for tradeoff functions in Theorem A.8 of the Appendix. $\square$

### 4.6  No multivariate CND for $f_{\epsilon,0}$

By the construction of Proposition 2.4, we know that a one-dimensional CND exists for any nontrivial tradeoff function. It turns out that the same cannot be said for the multivariate setting. In Theorem 4.8, we show that there is *no multivariate CND for $f_{\epsilon,0}$ with respect to any norm*. In fact, we prove the stronger result that it is not even possible to satisfy properties 1 and 2 of Definition 4.1

**Theorem 4.8.** *Let $d \geq 2$ and let $\|\cdot\|$ be any norm on $\mathbb{R}^d$. Then for any $\epsilon > 0$, there is no random vector satisfying properties 1 and 2 of Definition 4.1 for $f_{\epsilon,0}$ with respect to the norm $\|\cdot\|$. In particular, there is no multivariate CND for $f_{\epsilon,0}$.*

*Proof Sketch.* Suppose to the contrary, then $(\epsilon, 0)$-DP imposes strict bounds on the likelihood ratio of the distribution. These bounds allow us to find an arbitrarily long sequence of points, sufficiently far apart, where the density is bounded below. This ultimately shows that the density is not integrable. $\square$

Combining Theorem 4.8 with Proposition 4.2, we infer in Corollary 4.9 that $f_{\epsilon,0}$ cannot be written as the tensor product of any two nontrivial tradeoff functions. This means that if we want to design two independent mechanisms such that the joint release exactly satisfies $(\epsilon, 0)$-DP, then one of the mechanisms must be perfectly private.

**Corollary 4.9.** *Let $\epsilon > 0$ be given. There does not exist nontrivial symmetric tradeoff functions $f_1$ and $f_2$ such that $f_{\epsilon,0} = f_1 \otimes f_2$.*

**Remark 4.10.** Theorem 4.8 along with Theorem 4.3 gives an alternative argument that $f_{\epsilon,0}$ is not log-concave/infinitely decomposable.

## 5  Discussion

Motivated by the goals of constructing log-concave CNDs and multivariate CNDs, we found some fundamental connections between these constructions and the operations of mechanism composition and functional composition of the tradeoff functions. Surprisingly, the constructions for both log-concave and multivariate CNDs relied on whether a tradeoff function could be decomposed either according to functional composition, or according to mechanism composition. An interesting result of our work was that for $(\epsilon, 0)$-DP there is a unique 1-dimensional CND and no multidimensional CNDs, which implies that $f_{\epsilon,0}$ can neither be decomposed according to functional composition or mechanism composition. This highlights the limitations of pure-DP as a privacy definition. On the other hand, Gaussian-DP, Laplace-DP, and $(0, \delta)$-DP were seen to have much better properties.

While the framework of GDP and related notions (e.g., zero-concentrated DP) have many desirable properties, including those developed in this paper, there are still many reasons why one may be interested in other DP frameworks. In some applications, having a stronger notion of DP is needed to protect events with small probability, such as pure-DP or Laplace-DP; in this case, our work shows that Laplace-DP is a much better behaved notion of privacy than pure-DP. One may also propose other alternative DP definitions based on a family of tradeoff functions, and our research gives some

fundamental insights on what properties that family must have in order for log-concave or multivariate CNDs to be constructed.

We showed that a multivariate extension of CND can capture the same properties as in the 1-dimensional case. Awan and Vadhan [2021] showed that in one dimension, CNDs can be used to obtain DP hypothesis tests with optimal properties. An open question is whether our definition of a multivariate CND has any connections to optimal hypothesis testing.

Most of the constructions of multivariate CNDs presented in this paper are product distributions. Even the multivariate CNDs for GDP are a linear transformation of i.i.d. random variables. The $\ell_\infty$-mechanism is the exception, providing a truly nontrivial CND for Laplace-DP. It is worth exploring whether there are general techniques to produce nontrivial multivariate CNDs like the $\ell_\infty$-mechansism, as well as exploring the merits of such CNDs.

While many of the multivariate CNDs constructed for tradeoff functions, only held for specific sensitivity norms, a more general question would be on the existence and construction of multivariate CNDs for an arbitrary tradeoff/norm pair.

## Acknowledgments

This work was supported in part by NSF SES 2150615, awarded to Purdue University.

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
