# A    Appendix to Log-Concave and Multivariate Canonical Noise Distributions for Differential Privacy

## A.1    Broader impacts

Privacy is an important societal problem, and there is a natural tradeoff between the privacy afforded to the individuals of the dataset, and the utility of the published result. One may be concerned that differential privacy techniques reduce the utility of the results too much, in exchange for the privacy protection. In our work, by providing a better understanding of differential privacy, and by developing new mechanisms to achieve differential privacy, we make it possible to achieve higher utility at the same privacy cost; or alternatively, we can maintain the same utility while giving a stronger privacy protection. In our view, optimizing the privacy-utility tradeoff is universally beneficial to society, and we do not foresee any negative societal impacts of this work.

## A.2    Relations between functional composition and tensor product

Both the functional composition and the tensor product of tradeoff functions are essential concepts in our constructions of CNDs. In the remainder of this section, we establish some new relations between the two operations, which provide an interesting insight into the connection between group privacy and composition. First, we recall a lemma from Dong et al. [2022]:

**Lemma A.1** (Lemma A.5: Dong et al. [2022]). *Suppose that $T(P, Q) \geq f$ and $T(Q, R) \geq g$. Then $T(P, R) \geq g \circ f$.*

**Lemma A.2.** *Let $f$ and $g$ be any two symmetric tradeoff functions. Then $f \otimes g \geq f \circ g$.*

*Proof.* First note that if either $f$ or $g$ is equal to Id, then the result is trivial. Now, suppose that both $f$ and $g$ are nontrivial, and let $N_1 \sim F$ and $N_2 \sim G$ be independent, where $F$ is a CND for $f$ and $G$ is a CND for $g$.

By definition of the tensor product of tradeoff functions [Dong et al., 2022, Definition 3.1], we have that

$$T\left(\begin{pmatrix} 0 + N_1 \\ 0 + N_2 \end{pmatrix}, \begin{pmatrix} 1 + N_1 \\ 1 + N_2 \end{pmatrix}\right) = f \otimes g, \tag{1}$$

since $T(0 + N_1, 1 + N_1) = f$ and $T(0 + N_2, 1 + N_2) = g$, by definition of CND.

It is also true that

$$T\left(\begin{pmatrix} 0 + N_1 \\ 0 + N_2 \end{pmatrix}, \begin{pmatrix} 1 + N_1 \\ 0 + N_2 \end{pmatrix}\right) = f,$$

$$T\left(\begin{pmatrix} 1 + N_1 \\ 0 + N_2 \end{pmatrix}, \begin{pmatrix} 1 + N_1 \\ 1 + N_2 \end{pmatrix}\right) = g.$$

Then by Lemma A.1,

$$T\left(\begin{pmatrix} 0 + N_1 \\ 0 + N_2 \end{pmatrix}, \begin{pmatrix} 1 + N_1 \\ 1 + N_2 \end{pmatrix}\right) \geq g \circ f = f \circ g, \tag{2}$$

where the last equality follows since $f$ and $g$ are symmetric, using [Dong et al., 2022, Lemma A.4]. Comparing Equations (1) and (2), we have that $f \otimes g \geq f \circ g$. $\square$

**Remark A.3.** As a special case of Lemma A.2, we have that $f \otimes f \geq f \circ g$, which has an interesting interpretation: Suppose two situations: 1) your data is present once in two databases, and an $f$-DP mechanism is applied to each database once. This gives $f \otimes f$-DP cumulative privacy cost to you. 2) your data is present twice in one database, and an $f$-DP mechanism is applied once to the database. Since your data is present twice, by group privacy the incurred privacy cost to you is $f \circ f$-DP. Lemma A.2 says that you would prefer to be in the two separate databases. The intuition behind this can be understood as follows: in the second scenario, the privacy expert could choose to split the database into two: each one containing a copy of your data, and apply an $f$-DP mechanism to both. The nominal privacy cost of this would be $f$-DP (considering groups of size 1), as changing one entry affects only one of the two calculations. However, for groups of size two, the privacy cost is $f \circ f$-DP. This shows that all of the mechanisms in scenario 1 could also be applied to scenario

2, but in general there are mechanisms in scenario two that are not possible in scenario 1 (since in scenario 1, the databases cannot be merged).

Before we move on, we give a Lemma, extending Awan and Vadhan [2021, Lemma E.8] to arbitrary $k$. Lemma A.5 shows that given a CND $N$ for $f$, we can easily produce a CND for $f^{\circ k}$ by rescaling $N$ by $\frac{1}{k}$. To establish Lemma A.5, we need another technical lemma, which appeared within the proof Awan and Vadhan [2021, Lemma E.12]. We say that a cdf $F$ is *invertible* at $t$ if $F^{-1}(F(t)) = t$.

**Lemma A.4** (Awan and Vadhan [2021])**.** *Let $f$ be a nontrivial symmetric tradeoff function, and let $F$ be a CND for $f$. Call $M := \inf\{t \mid 0 < F(t)\}$. Then if $M < -1/2$ and $\alpha > 1 - f(1)$, then $F$ is invertible at $F^{-1}(\alpha) - 1$.*

*Proof.* Let $\alpha > 1 - f(1)$, or equivalently $1 - \alpha < f(1)$. Note that $F$ is invertible at $F^{-1}(\alpha)$, and by Awan and Vadhan [2021, Lemma E.3] $F$ is also invertible at $F^{-1}(\alpha) - 1$ unless $F^{-1}(\alpha) - 1 < M$. So, we need to show that $F^{-1}(\alpha) \geq M + 1$:

$$
\begin{align}
M + 1 &= \inf\{t + 1 \mid 0 < F(t)\} \tag{3} \\
&= \inf\{t \mid 0 < F(t - 1)\} \tag{4} \\
&= \inf\{t \mid 1 > 1 - F(t - 1)\} \tag{5} \\
&= \inf\{t \mid 1 - f(1) < 1 - f(1 - F(t - 1)) \;\&\; 0 < F(t - 1)\} \tag{6} \\
&= \inf\{t \mid 1 - f(1) < F(t) \;\&\; 0 < F(t - 1)\}, \tag{7}
\end{align}
$$

where (6) uses the fact that $1 - f$ is strictly decreasing at 1; (7) uses the fact that $0 < F(t - 1)$ to apply the recursion of Lemma 2.3. Now, suppose that $F(t - 1) = 0$: then $t - 1 \leq M$ and because $M < -1/2$, $F(t) < 1$. So, $0 = F(t - 1)$ implies that $0 = f(F(t))$. But this in turn implies that $F(t) \leq 1 - f(1)$. We see that $1 - f(1) < F(t)$ implies that $0 < F(t - 1)$. So,

$$
\begin{align}
M + 1 &= \inf\{t \mid 1 - f(1) < F(t)\} \tag{8} \\
&\leq \inf\{t \mid \alpha \leq F(t)\} \tag{9} \\
&= F^{-1}(\alpha), \tag{10}
\end{align}
$$

where (9) uses the fact that $\alpha > 1 - f(1)$. We see that $F^{-1}(\alpha) - 1 \geq M$ and conclude that $F$ is invertible at $F^{-1}(\alpha) - 1$. $\qquad\square$

**Lemma A.5.** *Let $F$ be a CND for a nontrivial symmetric tradeoff function $f$. Then $F(k\cdot)$ is a CND for $f^{\circ k}$ for any $k \in \mathbb{N}^+$.*

*Proof.* For any $k$, denote $F_k(x) = F(kx)$ and $F_k^{-1}(x) = \frac{1}{k} F^{-1}(x)$, where $F_k^{-1}$ is the quantile function of $F_k$. Symmetry and continuity of $F_k$ are clear.

For induction, assume that for some $k > 1$, we have that $F_{k-1}$ is a CND for $f^{\circ(k-1)}$. In particular, we have that

$$
\begin{align*}
f^{\circ(k-1)} &= F_{k-1}(F_{k-1}^{-1}(\alpha) - 1) \\
&= F[(k - 1)\{[1/(k - 1)]F^{-1}(\alpha) - 1\}] \\
&= F(F^{-1}(\alpha) - (k - 1)).
\end{align*}
$$

Let $M := \inf\{t \mid 0 < F_{k-1}(t)\}$. By symmetry of $F_{k-1}$, we know that $M \leq 0$. If $M \geq -1/2$, then we have that $f^{\circ k}(\alpha) \leq f^{\circ(k-1)}(\alpha) = F_{k-1}(F_{k-1}^{-1}(\alpha) - 1) = 0$ for all $\alpha \in (0, 1)$; we also have $F_k(F_k^{-1}(\alpha) - 1) = F(F^{-1}(\alpha) - k) = 0 = f^{\circ k}(\alpha)$. Furthermore, $T(F_k(\cdot), F_k(\cdot - 1)) = T(F(\cdot), F(\cdot - k)) = 0 = g$, since $F(\cdot)$ and $F(\cdot - k)$ have disjoint support. Finally, note that $T(F_k(\cdot), F_k(\cdot - m)) \geq 0 = T(F_k(\cdot), F_k(\cdot - 1))$, since 0 is a trivial lower bound for any tradeoff function. We conclude that when $M \geq -1/2$, $F_k$ is a CND for $f^{\circ k}$.

Now suppose that $M < -1/2$ and let $\alpha \in (0, 1)$. If $\alpha \leq 1 - f^{\circ(k-1)}(1)$, then $f^{\circ k}(\alpha) = f(f^{\circ(k-1)}(\alpha)) = f(0) = 0$ and $F(F^{-1}(1 - \alpha) - k) \leq F(F^{-1}(1 - \alpha) - (k - 1)) = f^{\circ(k-1)}(\alpha) = $

$0 = f^{\circ k}(\alpha)$ because $F$ is increasing. We see that $f^{\circ k} = F_k(F_k^{-1}(\alpha) - 1)$ in this case. Now assume that $\alpha > 1 - f(1)$. Then by Lemma A.4, we have that $F_{k-1}$ is invertible at $F_{k-1}^{-1}(\alpha) - 1$. Then,

$$f^{\circ k} = f \circ F_{k-1}(F_{k-1}^{-1}(\alpha) - 1) \tag{11}$$

$$= F(F^{-1}[F_{k-1}(F_{k-1}^{-1}(\alpha) - 1)] - 1) \tag{12}$$

$$= F((k-1)F_{k-1}^{-1}[F_{k-1}(F_{k-1}^{-1}(\alpha) - 1)] - 1) \tag{13}$$

$$= F((k-1)(F_{k-1}^{-1}(\alpha) - 1) - 1) \tag{14}$$

$$= F(F^{-1}(\alpha) - k) \tag{15}$$

$$= F_k(F_k^{-1}(\alpha) - 1), \tag{16}$$

where in (14), we used the fact that $F_{k-1}$ is invertible at $F_{k-1}^{-1}(\alpha) - 1$.

We have shown that $f^{\circ k} = F_k(F_k^{-1}(\alpha) - 1)$. Since $F_k(F_k^{-1}(\alpha) - 1)$ represents the type II error of the (potentially suboptimal) test, which rejects when the observed random variable is above a certain threshold, we have that $T(N_k, N_k + 1) = T(N, N + k) \leq f^{\circ k}$, where $N \sim F$ and $N_k \sim F_k$. To verify properties 2 and 3 of Definition 2.2, it remains to show that $T(N_k, N_k + 1) \geq f^{\circ k}$. Note that $T(N, N + (k-1)) = f^{\circ(k-1)}$, and $T(N + (k-1), N + k) = T(N, N + 1) = f$. By Lemma A.1, we have that $T(N_k, N_k + 1) = T(N, N + k) \geq f^{\circ(k-1)} \circ f = f^{\circ k}$, which completes the argument for parts 2 and 3 of Definition 2.2.

For property 1 of Definition 2.2, let $m \in [0, 1]$. As before, we use the notation $N \sim F$, $N_k \sim F_k$ and $N_{k-1} \sim F_{k-1}$. We will show that $T(N_k, N_k + m) = T(N, N + km) \geq f^{\circ k}$. If $m \leq \frac{k-1}{k}$, then $m^* = \frac{km}{k-1} \in [0, 1]$. In this case,

$$T(N, N + km) = T\left(N, N + (k-1)\frac{km}{k-1}\right)$$
$$= T(N_{k-1}, N_{k-1} + m^*)$$
$$\geq T(N_{k-1}, N_{k-1} + 1)$$
$$= f^{\circ(k-1)}$$
$$\geq f^{\circ k},$$

where we used the fact that $F_{k-1}$ is a CND for $f^{\circ(k-1)}$ and that $f^{\circ(k-1)} \geq f^{\circ k}$. If $m \geq \frac{k-1}{k}$, then $m^* = km - (k-1) \in [0, 1]$. Then

$$T(N, N + km) = T(N, N + (k-1) + m^*) \tag{17}$$

$$\geq T(N, N + (k-1)) \circ T(N, N + m^*) \tag{18}$$

$$= f^{\circ(k-1)} \circ T(N, N + m^*) \tag{19}$$

$$\geq f^{\circ(k-1)} \circ f \tag{20}$$

$$= f^{\circ k}, \tag{21}$$

where for (18) we use Lemma A.1 and the fact that $T(N+(k-1), N+(k-1)+m^*) = T(N, N+m^*)$, and for (20), we use the inductive hypothesis that $F_{k-1}$ is a CND for $f^{\circ(k-1)}$. $\square$

**Example A.6** (Composition and Group Privacy do not Commute). It is an interesting question whether the following property holds: $(f \otimes g)^{\circ k} = f^{\circ k} \otimes g^{\circ k}$. This is true for GDP:

$$(G_{\mu_1} \otimes G_{\mu_2})^{\circ k} = G_{k\|\binom{\mu_1}{\mu_2}\|} = G_{\|\binom{k\mu_1}{k\mu_2}\|} = G_{\mu_1}^{\circ k} \otimes G_{\mu_2}^{\circ k}.$$

However, by studying $(0, \delta)$-DP, we see that this property does not hold in general – even for log-concave tradeoff functions. We compute that

$$(f_{0,\delta_1} \otimes f_{0,\delta_2})^{\circ k} = f_{0,1-(1-\delta_1)(1-\delta_2)}^{\circ k} = f_{0,\min\{1,k(1-(1-\delta_1)(1-\delta_2))\}},$$

whereas $f_{0,\delta_1}^{\circ k} \otimes f_{0,\delta_2}^{\circ k} = f_{0,\min\{1,k\delta_1\}} \otimes f_{0,\min\{1,k\delta_2\}} = f_{0,1-(1-\min\{1,k\delta_1\})(1-\min\{1,k\delta_2\})}.$ plugging in $k = 2$ and $\delta_1 = \delta_2 = .1$, we get that the first expression yields .38, whereas the second gives .36. Interestingly, it seems that accounting for group privacy *first*, before applying composition gives the tighter privacy analysis. This is confirmed by the inequality in Proposition A.7.

**Proposition A.7.** *Let $f$ and $g$ be tradeoff functions. Then $(f \otimes g)^{\circ k} \leq f^{\circ k} \otimes g^{\circ k}$.*

*Proof.* We know that, $f = T(X, X+1)$ and $g = T(Y, Y+1)$, where $X$ is a CND for $f$ and $Y$ is a CND for $g$.

$$
\begin{aligned}
(f \otimes g)^{\circ k} &= \big(T[X, X+1] \otimes T[Y, Y+1]\big)^{\circ k} \\
&= \big(T[(X, Y), (X+1, Y+1)]\big)^{\circ k} \\
&\leq T[(X, Y), (X+k, Y+k)] \qquad \text{by Lemma A.1 and Lemma A.5} \\
&= T[X, X+k] \otimes T[Y, Y+k] \\
&= T[X/k, X/k+1] \otimes T[Y/k, Y/k+1] \\
&= f^{\circ k} \otimes g^{\circ k},
\end{aligned}
$$

since $X/k$ and $Y/k$ are CNDs for $f^{\circ k}$ and $g^{\circ k}$ respectively, by Lemma A.5. $\qquad \square$

### A.3 A limit theorem for tradeoff functions

Below, we introduce a limit theorem for tradeoff functions, which can be used to show a mechanism satisfies $f$-DP in terms of certain limits.

**Theorem A.8.** *Let $P_n \overset{\mathrm{TV}}{\to} P$ and $Q_n \overset{\mathrm{TV}}{\to} Q$ be two sequences of distributions, which converge in total variation. Then $T(P_n, Q_n) \to T(P, Q)$ uniformly.*

*Proof.* By Dong et al. [2022, Lemma A.7], it suffices to prove point-wise convergence. First we will establish $T(P, Q)$ as an asymptotic lower bound on $T(P_n, Q_n)$. By Lemma A.1, we have that

$$
T(P_n, Q_n) \geq T(Q, Q_n) \circ T(P, Q) \circ T(P_n, P).
$$

Since $P_n \overset{\mathrm{TV}}{\to} P$ and $Q_n \overset{\mathrm{TV}}{\to} Q$, we have that $T(P_n, P)(\alpha) \geq [(\alpha - TV(P_n, P))]_0^1$ and $T(Q_n, Q)(\alpha) \geq [(\alpha - TV(Q_n, Q))]_0^1$, where $[x]_a^b := \max\{\min\{x, b\}, a\}$ is the clamping function. Since all tradeoff functions are increasing, the following inequality holds:

$$
T(P_n, Q_n) \geq [(\alpha - \mathrm{TV}(Q_n, Q))]_0^1 \circ T(P, Q) \circ [(\alpha - \mathrm{TV}(P_n, P))]_0^1 \to T(P, Q),
$$

and the limit holds since $\mathrm{TV}(P_n, P) \to \mathrm{Id}$, $\mathrm{TV}(Q_n, Q) \to \mathrm{Id}$, and tradeoff functions are continuous.

Next, we show that $T(P, Q)$ is an asymptotic upper bound for $T(P_n, Q_n)$. It suffices to check for $\alpha \in (0, 1)$, since tradeoff functions are continuous. Let $\alpha^* \in (0, 1)$ be given. Let $\phi$ be an optimal test for $T(P, Q)$ such that $\mathbb{E}_P \phi = \alpha^*$ and $\mathbb{E}_Q \phi = 1 - f(1 - \alpha^*)$. Note that if $U \sim U(0, 1)$, we can write

$$
\mathbb{E}_P \phi = \mathbb{E}_{X \sim P, U} I(U \leq \phi(X)) = \mathbb{E}_U P_{X \sim P}(U \leq \phi(X) | U) = \mathbb{E}_U P(\phi^{-1}([U, 1]) | U),
$$

which will allow us to apply the total variation convergence. Call $\alpha_n = \mathbb{E}_{P_n} \phi$ for all $n$. Notice that $\alpha_n \to \alpha^*$, since

$$
\begin{aligned}
|\alpha_n - \alpha^*| &= |\mathbb{E}_{P_n} \phi - \mathbb{E}_P \phi| \\
&= \left| \mathbb{E}_U P_n(\phi^{-1}([U, 1]) \mid U) - \mathbb{E}_U P(\phi^{-1}([U, 1]) \mid U) \right| \\
&\leq \mathbb{E}_U \left| P_n(\phi^{-1}([U, 1]) \mid U) - P(\phi^{-1}([U, 1]) \mid U) \right| \\
&\leq \mathbb{E}_U \mathrm{TV}(P_n, P) \\
&\to 0,
\end{aligned}
$$

as $P_n \overset{\mathrm{TV}}{\to} P$. Similarly, we have that $|\mathbb{E}_{Q_n} \phi - \mathbb{E}_Q \phi| \to 0$, implying that $\mathbb{E}_{Q_n} \phi \to 1 - f(1 - \alpha^*)$. Then,

$$
\begin{aligned}
T(P_n, Q_n)(\alpha_n) &\leq 1 - \mathbb{E}_{Q_n} \phi \\
&\to f(1 - \alpha^*) \\
&= T(P, Q)(\alpha^*).
\end{aligned}
$$

However, we actually want to show that $T(P_n, Q_n)(\alpha^*)$ is asymptotically upper bounded by $T(P, Q)(\alpha)$. Luckily, $T(P_n, Q_n)(\alpha_n)$ and $T(P_n, Q_n)(\alpha^*)$ are close for large $n$, since tradeoff functions are "locally Lipschitz." We explain as follows: Since $\alpha_n \to \alpha^*$, let $N$ be such that for all $n \geq N$, $\alpha_n \in \left(0, \alpha^* + \frac{1-\alpha^*}{2}\right)$. On the interval $(0, \alpha^* + \frac{1-\alpha^*}{2})$, we claim that $T(P_n, Q_n)$ is $\frac{2}{1-\alpha^*}$-Lipschitz. This is because the derivative (or subderivative) of a convex function is increasing, and the tangent lines of a convex function are always a lower bound. In the worst case, using the points $(\alpha^* + (1 - \alpha^*)/2, 0)$ and $(1, 1)$, the slope at $\alpha^* + (1 - \alpha^*)/2$ is at most $\frac{1-0}{1-(\alpha^*+(1-\alpha^*)/2)} = \frac{2}{1-\alpha^*}$.
Now that we have established that $T(P_n, Q_n)$ is $\frac{2}{1-\alpha^*}$-Lipschitz on $(0, \alpha^* - (1 - \alpha^*)/2)$, we have that for all $n \geq 0$,

$$\left| T(P_n, Q_n)(\alpha_n) - T(P_n, Q_n)(\alpha^*) \right| \leq \frac{2}{1 - \alpha^*} |\alpha_n - \alpha| \to 0.$$

We conclude that $T(P_n, Q_n)(\alpha^*)$ is asymptotically upper bounded by $T(P, Q)(\alpha^*)$ for all $\alpha* \in (0, 1)$. Combining the asymptotic lower and upper bounds, we have that $T(P_n, Q_n) \to T(P, Q)$. □

Two immediate corollaries of the above theorem are as follows. The first, generally states that if we establish a lower bound on $T(P_n, Q_n)$ for all $n$, and $P_n \overset{\text{TV}}{\to} P$ and $Q_n \overset{\text{TV}}{\to} Q$, then the lower bound applies to $T(P, Q)$ as well. This could be generalized to a sequence of lower bounds $f_n \to f$ as well.

**Corollary A.9.** *Let $P_n \overset{\text{TV}}{\to} P$ and $Q_n \overset{\text{TV}}{\to} Q$ be two sequences of distributions such that $T(P_n, Q_n) \geq f$ for all $n$. Then $T(P, Q) \geq f$.*

Corollary A.10 shows that the limit of an $f$-DP mechanism satisfies $f$-DP (could also replace each $f$ with $f_n \to f$). This is similar to the limit result of Kifer et al. [2012], but is phrased in terms of convergence in total variation rather than almost sure convergence.

**Corollary A.10.** *Let $M_n$ be a sequence of mechanisms satisfying $f$-DP (i.e., $T(M_n(D), M_n(D')) \geq f$ for all adjacent $D$ and $D'$), and suppose that $M_n(D) \overset{\text{TV}}{\to} M(D)$ for all $D$. Then $M$ satisfies $f$-DP: $T(M(D), M(D')) \geq f$.*

### A.4 Proofs and technical lemmas for the main paper

For any measurable set $A$, let $\lambda(A)$ denote the Lebesgue measure of $A$.

**Lemma A.11.** *Let $A, B \subset [-1/2, 1/2]$ be disjoint sets with positive Lebesgue measure such that $A \cup B = [-1/2, 1/2]$. Then there exists a shift $\omega \in (-1, 1)$ such that $(B + \omega) \cap A$ has positive Lebesgue measure.*

*Proof.* Suppose to the contrary that for all $\omega \in (-1, 1)$, $\lambda((B + \omega) \cap A) = 0$. This implies that

$$0 = \int_{-1}^{1} \lambda((B + \omega) \cap A) \, d\omega \tag{22}$$

$$= \int_{-1}^{1} \int_{-1/2}^{1/2} I(x \in (B + \omega) \cap A) \, dx \, d\omega \tag{23}$$

$$= \int_{-1}^{1} \int_{A} I(x \in B + \omega) \, dx \, d\omega \tag{24}$$

$$(\text{Tonelli's Theorem}) = \int_{A} \int_{-1}^{1} I(x \in B + \omega) \, d\omega \, dx \tag{25}$$

$$= \int_{A} \lambda(x - B) \, dx \tag{26}$$

$$= \int_{A} \lambda(B) \, dx \tag{27}$$

$$= \lambda(A)\lambda(B), \tag{28}$$

where we used Tonelli's Theorem in (25) to change the order of integration, in (26) we used the fact $x - B \subset [-1, 1]$ since both $x$ and $B$ lie in $[-1/2, 1/2]$, and in (27) we used the fact that

Lebesgue measure is translation invariant. We see that either $\lambda(A) = 0$ or $\lambda(B) = 0$, giving a contradiction. $\qquad\square$

**Proposition 3.1.** *Let $\epsilon > 0$. The distribution* $\mathrm{Tulap}(0, \exp(-\epsilon), 0)$ *is the unique CND for $f_{\epsilon,0}$.*

*Proof.* Let $g$ be the density of an arbitrary CND for $f_{\epsilon,0}$, and let $G$ denote its cdf function. Since $g$ is the density of a symmetric random variable centered at zero, $g(x) = g(-x)$ for all $x \in \mathbb{R}$. By Awan and Vadhan [2021, Proposition 3.7], we have that $G(x+1) = 1 - f_{\epsilon,0}(1 - G(x))$, which implies that $g(x+1) = f'_{\epsilon,0}(1 - G(x))g(x)$. For $x > 0$, we have that $G(x) \geq 1/2 > c$, where $c$ satisfies $f_{\epsilon,0}(1 - c) = c$. Recall that $f_{\epsilon,0}(\alpha) = \alpha e^{-\epsilon}$ for all $\alpha \leq c$. Then $f'_{\epsilon,0}(1 - G(x)) = e^{-\epsilon}$ for $x \geq 0$. We see that we can write $g(u+k) = e^{-|k|\epsilon}g(u)$ for $u \in [-1/2, 1/2]$ and $k \in \mathbb{Z}$.

We see that so far, $g$ has the freedom to choose its values in $[-1/2, 1/2]$ and then all other values are determined by the above recurrence. Note that for the Tulap distribution, its density is the constant value of $\frac{\exp(\epsilon)-1}{\exp(\epsilon)+1}$ on $(-1/2, 1/2)$, since it is the constructed CND by Proposition 2.4 [Awan and Vadhan, 2021, Corollary 3.10]. Suppose that $g(u)$ is non constant on $(-1/2, 1/2)$. Then it must take on some values above and below $\frac{\exp(\epsilon)-1}{\exp(\epsilon)+1}$ in order to still integrate to 1. To rule out trivial cases, where $g(x)$ is equivalent to the Tulap density up to a set of measure zero, we assume that the sets

$$A := \left\{ u \in (-1/2, 1/2) \mid g(u) > \frac{\exp(-\epsilon)-1}{\exp(\epsilon)+1} \right\},$$

$$B := \left\{ u \in (-1/2, 1/2) \mid g(u) \leq \frac{\exp(-\epsilon)-1}{\exp(\epsilon)+1} \right\},$$

both have positive Lebesgue measure. We denote $\lambda$ as the Lebesgue measure.

By Lemma A.11, we know that there exists $\omega \in (-1, 1)$ such that $(B + \omega) \cap A$ has positive Lebesgue measure. By symmetry of $g$ about zero, there exists a positive shift $\omega \in (0, 1)$ such that $\lambda((B+\omega)\cap A) > 0$. Consider $\Delta := 1 - \omega \in (0, 1)$. Let $v \in (B+\omega)\cap A$ and $u := v - \omega \in B \cap (A-\omega)$, and consider the likelihood ratio:

$$\frac{g(1+u-\Delta)}{g(1+u)} = \frac{g(u+\omega)}{g(1+u)} = \frac{g(v)}{e^{-\epsilon}g(u)} > e^{\epsilon},$$

where we use the fact that $u \in B$, $v \in A$, and $\lambda((B+\omega)\cap A) > 0$. This means that the likelihood ratio $g(x-\Delta)/g(y)$ is not bounded by $\exp(\epsilon)$ almost everywhere, for all $\Delta \in [-1, 1]$; by Awan et al. [2019, Proposition 2.3], this means that the additive mechanism with density $g$ does not satisfy $\epsilon$-DP. In other words, for $X \sim g$ and $\Delta$ as above, $T(X, X + \Delta)(\alpha) < f_{\epsilon,0}(\alpha)$ for some $\alpha \in (0, 1)$. We conclude that $g$ is not a CND for $f_{\epsilon,0}$. The only assumption we made about $g$ was that it was non-constant on $(-1/2, 1/2)$ on a set of positive probability. Due to the contradiction, we conclude that $g$ is in fact constant on $[-1/2, 1/2]$ almost everywhere, which means that it is distributed as $\mathrm{Tulap}(0, \exp(-\epsilon), 0)$. $\qquad\square$

**Theorem 3.3.** *A nontrivial tradeoff function $f \in \mathscr{F}$ is log-concave if and only if it is infinitely divisible. In particular,*

1. *If $f$ is log-concave with log-concave CND $N \sim F$, then $\{f_t \mid t \geq 0\}$ defined by $f_t = F(F^{-1}(\alpha) - t)$ satisfies the assumptions of Definition 3.2.*

2. *Let $f$ be infinitely divisible, with monoid $\{f_t \in \mathscr{F} \mid t \geq 0\}$, as defined in Definition 3.2, such that $f = f_1$. Let $F_s$ be any CND for $f_s$ (such as constructed in Proposition 2.4). Then the following limit exists $F^*(t) := \lim_{s\to 0} F_s(\frac{1}{s}t)$ and $N \sim F^*$ is the unique log-concave CND for $f$. Furthermore, $F^*(st)$ is the unique log-concave CND for $f_s$, for all $s > 0$.*

*Proof.* 1) Let $f$ be a log-concave tradeoff function with log-concave CND $F$. Define $f_s(\alpha) = F(F^{-1}(\alpha) - s)$, which is a tradeoff function since $F$ is log-concave. Note that $f_s \circ f_t = f_{s+t}$, $f_t$ is nontrivial except wen $t = 0$, and $f_t \to \mathrm{Id}$ as $t \to 0$.

2) For part 2, we first show that the limit exists for the specific sequence $s_n = 1/n!$, and then we will show that convergence holds for all sequences that converge to zero. By construction, $F_{s_n}(\cdot)$ is a CND for $f_{s_n}$. So, $F_{s_n}(\cdot)$ has values determined on $(1/2)\mathbb{Z}$, no matter the choice of CND. Furthermore,

$F_{s_n}(t/s_n)$ is a CND for $f_1 = f_{s_n}^{\circ n!}$, by Lemma A.5. Then for any choice of CND for $F_{s_n}$, the cdf $F_{s_n}(t/s_n)$ has values determined on $(s_n/2)\mathbb{Z}$, and $F_{s_n}(t/s_n)$ is a continuous cdf (as it is a CND). Note that the sequence $(s_n)_{n=1}^{\infty}$ satisfies $(s_n/2)\mathbb{Z} \subset (s_{n+1}/2)\mathbb{Z}$ for all $n$, and as $n \to \infty$, we have that $\bigcup_{n=1}^{\infty}((s_n/2)\mathbb{Z}) = \mathbb{Q}$, the set of rational numbers. In words, the set of determined values of $F_{s_n}(t/s_n)$ is an increasing sequence of sets, whose limit is the rational numbers. Due to this, the sequence of CNDs $F_{s_n}(t/s_n)$ is "pinned down" at an increasing number of points, and is eventually determined at every rational number. Since every $F_{s_n}(t/s_n)$ is monotone, and $\mathbb{Q}$ is dense in $\mathbb{R}$, it follows that the limit of this sequence, $F^*$, is a unique monotone function.

Next we show that $F^*$ is a continuous cdf. We already mentioned that $F^*$ is non-decreasing, and it is easy to show that $\lim_{t \to +\infty} F^*(t) = 1$ and $\lim_{t \to -\infty} F^*(t) = 0$. The challenging part is to show that $F^*$ is continuous. It suffices to show that the convergence of $(F_{s_n}(t/s_n))_{n=1}^{\infty}$ to $F^*$ is uniform. Before we show this, we establish the following inequality: for all $t \in \mathbb{R}$, $|F_{s_n}(t/s_n) - F^*(t)| \leq \sup_t |t - f_{s_n}(t)|$. To see this, let $t \in \mathbb{R}$. Then for each $n$, there exists $k_n \in \mathbb{Z}$ such that $\frac{(k_n-1)s_n}{2} \leq t \leq \frac{(k_n+1)s_n}{2}$. Since $F^*$ is a non-decreasing function, this implies that

$$F^*\left(\frac{(k_n-1)s_n}{2}\right) \leq F^*(t) \leq F^*\left(\frac{(k_n+1)s_n}{2}\right)$$

$$F_{s_n}\left(\frac{k_n-1}{2}\right) \leq F^*(t) \leq F_{s_n}\left(\frac{k_n+1}{2}\right)$$

$$f_{s_n}\left(F_{s_n}\left(\frac{k_n+1}{2}\right)\right) \leq F^*(t) \leq F_{s_n}\left(\frac{k_n+1}{2}\right),$$

where if $F_{s_n}(\frac{k_n+1}{2}) < 1$ the third line is equivalent to the second line by Lemma 2.3, and if $F_{s_n}(\frac{k_n+1}{2}) = 1$, then the inequality in the third line is potentially weaker. By similar reasoning, we have that $f_{s_n}(F_{s_n}(\frac{k_n+1}{2})) \leq F_{s_n}(t/s_n) \leq F_{s_n}(\frac{k_n+1}{2})$ as well. Therefore,

$$|F^*(t) - F_{s_n}(t/s_n)| \leq \left|F_{s_n}\left(\frac{k_n+1}{2}\right) - f_{s_n}\left(F_{s_n}\left(\frac{k_n+1}{2}\right)\right)\right| \leq \sup_{t \in [0,1]} |t - f_{s_n}(t)|.$$

We are now ready to prove uniform convergence. Let $\epsilon > 0$ be given. Let $N \in \mathbb{Z}^+$ be such that $\sup_{t \in [0,1]} |t - f_{s_n}(t)| < \epsilon$, which is possible since $f_{s_n}(t) \to t$ uniformly (Polya's theorem). Then for all $n \geq N$, we have that $|F_{s_n}(t/s_n) - F^*(t)| \leq \sup_t |t - f_{s_n}(t)| < \epsilon$. Uniform convergence of continuous functions implies that the limit function is also continuous, so we conclude that $F^*$ is a continuous cdf.

Next we will show that for all $t \in \mathbb{R}^+$, $f_t = F^*(F^{*-1}(\alpha) - t)$. Let $(-x, x) := F^{*-1}((0,1))$ be the support of the distribution $F^*$. Let $q \in \mathbb{Q}^+$ be the "shift," and let $p \in (-\infty, x - q) \cap \mathbb{Q}$ be the "threshold" in the test. Let $n \in \mathbb{Z}^+$ be such that $p = a/n!$ and $q = b/n!$ for some $a \in \mathbb{Z}$ and $b \in \mathbb{Z}^+$. As we did earlier, denote $s_n = 1/n!$. Recall that $F^*(t)$ and $F_{s_n}(t/s_n)$ agree for all $t \in \frac{s_n}{2}\mathbb{Z}$. In particular, $p \in \frac{s_n}{2}\mathbb{Z}$. Then

$$F^*(p) = F^*(s_n a)$$
$$= F_{s_n}(a)$$
$$= f_{s_n} \circ F_{s_n}(a+1)$$
$$= f_{s_n}^{\circ b} \circ F_{s_n}(a+b)$$
$$= f_q \circ F_{s_n}(a+b)$$
$$= f_q \circ F^*(s_n(a+b))$$
$$= f_q \circ F^*(p+q),$$

where we used the fact that $p < x - q$ to establish that $F_{s_n}(a+b) = F^*(p+q) < 1$, which enabled the recurrence application of Lemma 2.3. Furthermore, since the rational numbers are dense in $\mathbb{R}$, $F^*$ is continuous, and $f_q$ is continuous on $[0,1)$, we have that $F^*(t) = f_q \circ F^*(t+q)$ for all $t$ such that $t < x - q$. Now let $\alpha \in [0,1)$ and call $t_\alpha = F^{*-1}(\alpha) - q$. Note that $F^*(q + t_\alpha) = \alpha$ and that $t_\alpha < x - q$. Then we have that $F^*(F^{*-1}(\alpha) - q) = f_q(\alpha)$ for all $\alpha \in [0,1)$. Finally, we extend the result for arbitrary $f_r$, $r \in \mathbb{R}^+$. Let $q_n \in \mathbb{Q}^+$ be a sequence such that $q_n \to r$. Then

$$f_r = f_{r-q_n} \circ f_{q_n} = f_{r-q_n} \circ F^*(F^{*-1}(\alpha) - q_n) \to \text{Id} \circ F^*(F^{*-1}(\alpha) - r),$$

since $f_{r-q_n}$ converges to Id uniformly, and $F^*$ is continuous. We have that $F^*(s\cdot)$ is a CND for $f_s$: the symmetry of $F^*$ is obvious, and the fact that $F^*$ satisfies DP follows by the property that $f_s = F^*(F^{*-1}(\alpha) - s)$.

Let $s > 0$. Let $N \sim F^*$. Since $N/s \sim F^*(st)$ is a CND for $f_s$, we have that $T(N, N+s)(\alpha) = T(N/s, N/s+1)(\alpha) = f_s(\alpha) = F^*(F^{*-1}(\alpha) - s)$. However, $T(N, N+s)(\alpha) = F^*(F^{*-1}(\alpha) - s)$ holds for all $s > 0$ if and only if $F^*$ has a log-concave density [Dong et al., 2022, Lemma A.3]. Therefore $F^*$ is a log-concave distribution, and $F^*(s\cdot)$ is a log-concave CND for $f_s$ for all $s > 0$.

Finally, we will make sure the limit does not depend on the specific sequence $s_n = 1/n!$. We will use a very similar argument as when we established uniform convergence to show that for any positive sequence $r_n$ which converges to zero, $F_{r_n}(t/r_n)$ also converges uniformly to $F^*(t)$. Let $t \in \mathbb{R}$. Then for any $n \in \mathbb{Z}^+$, there exists $k_n$ such that $\frac{(k_n-1)r_n}{2} \leq t \leq \frac{(k+1)r_n}{2}$. Then

$$F^*\left(\frac{(k_n-1)r_n}{2}\right) \leq F^*(t) \leq F^*\left(\frac{(k_n+1)r_n}{2}\right).$$

Since $F^*(\cdot r_n)$ and $F_{r_n}(\cdot)$ are both CNDs for $f_{r_n}$, they agree on all half integer values. So,

$$F_{r_n}\left(\frac{k_n-1}{2}\right) \leq F^*(t) \leq F_{r_n}\left(\frac{k_n+1}{2}\right)$$

$$f_{r_n}\left(F_{r_n}\left(\frac{k+1}{2}\right)\right) \leq F^*(t) \leq F_{r_n}\left(\frac{k_n+1}{2}\right).$$

By similar reasoning, we have that $f_{r_n}\left(F_{r_n}\left(\frac{k+1}{2}\right)\right) \leq F_{r_n}(t/r_n) \leq F_{r_n}\left(\frac{k_n+1}{2}\right)$. Then

$$|F^*(t) - F_{r_n}(t/r_n)| \leq \left|F_{r_n}\left(\frac{k_n+1}{2}\right) - f_{r_n}\left(F_{r_n}\left(\frac{k_n+1}{2}\right)\right)\right| \leq \sup_{t\in[0,1]} |t - f_{r_n}(t)|.$$

Since $r_n \to 0$, we have that $f_{r_n}(t)$ converges uniformly to $t$. So, we have that $F_{r_n}(t/r_n)$ converges uniformly to $F^*(t)$. $\qquad\square$

**Lemma A.12.** *Let $f$ and $g$ be tradeoff functions.*

1. *If $f$ and $g$ are piece-wise linear with $k$ and $\ell$ break points (respectively), and $f$ satisfies $f(x) = 0$ implies $x = 0$, then $f \circ g$ is piece-wise linear with at most $k + \ell$ break points, and at least $\max\{k, \ell\}$ break points.*

2. *If $f$ is piece-wise linear with $k \geq 1$ break points, and $f(x) = 0$ implies that $x = 0$, then $f^{\circ n}$ has at least $k + (n-1)$ break points.*

3. *If $f \circ g$ is piece-wise linear, then $g$ is piece-wise linear on $[0, 1]$ and $f$ is piece-wise linear on $[0, g(1)]$. (note that $f$ can be arbitrary on $(g(1), 1]$ and it does not affect $f \circ g$)*

*Proof.*   1. The composition of linear functions is linear. So, it is clear that $f \circ g$ is piece-wise linear. Let $B_g$ be the set of break points of $g$ and $B_f$ be the set of break points of $f$. Then the break points of $f \circ g$ are $g^{-1}(B_f) \bigcup B_g$, since $f$ is invertible, which has at most $|B_f| + |B_g| = k + \ell$ elements, and at least $\max\{k, \ell\}$ elements.

2. Let $B_f$ be the set of break points of $f$. Then the set of break points of $f^{\circ n}$ is $B_f \cup f^{-1}(B_f) \cup f^{-1}(f^{-1})(B_f) \cup \cdots \cup (f^{-1})^{\circ(n-1)}(B_f)$. The number of break points of $f^{\circ n}$ is then lower bounded by $|B_f \cup f(B_f) \cup f^{\circ 2}(B_f) \cup \cdots \cup f^{\circ(n-1)}(B_f)|$, by applying $f^{\circ(n-1)}$ to each of the sets (since applying a function to a set cannot increase the cardinality). We know that $|B_f| = k$. Because $f(x) = 0$ implies that $x = 0$, we have that $f$ is strictly increasing on $(0, 1)$; so we have that $|f(B_f)| = k$ as well. Furthermore, for each $x \in B_f$, $f(x) < x$ as $f$ is nontrivial ($k \geq 1$ implies nontrivial). Let $x_m$ be the minimum element in $B_f$. Then $f(x_m) \in f(B_f)$ and $f(x_m) \notin B_f$. So, $|B_f \cup f(B_f)| \geq |B_f| + 1 = k+1$. Continuing this process, we get that the number of break points of $f^{\circ n}$ is at least $k + n - 1$.

3. Since $f \circ g$ is piece-wise linear, $\frac{d^2}{d\alpha^2} f \circ g(\alpha) = 0$ except at finitely many values. Then

$$
\begin{aligned}
0 &= \frac{d^2}{d\alpha^2} f \circ g(\alpha) \\
&= \frac{d}{d\alpha} \left( f'(g(\alpha)) g'(\alpha) \right) \\
&= f''(g(\alpha))(g'(\alpha))^2 + f'(g(\alpha)) g''(\alpha),
\end{aligned}
$$

except at finitely many values. Note that since $f$ and $g$ are non-decreasing and convex, the following quantities are non-negative (whenever they are well-defined): $f'$, $g'$, $f''$, and $g''$. So, the above equation implies that for all but finitely many $\alpha$, either $f''(g(\alpha)) = 0$ or $g'(\alpha) = 0$ and either $f'(g(\alpha)) = 0$ or $g''(\alpha) = 0$. Note that $g$ is zero on $\{\alpha \mid g'(\alpha) = 0\}$ and $f$ is zero on $\{g(\alpha) \mid f'(g(\alpha)) = 0\}$. Furthermore, $g$ is piece-wise linear on $\{\alpha \mid g''(\alpha) = 0\}$ and $f$ is piece-wise linear on $\{g(\alpha) \mid f''(g(\alpha)) = 0\}$. We see that on $(0, 1)$, $g$ is either zero or piece-wise linear, and so it is piece-wise linear on $[0, 1]$. Similarly on $(g(0), g(1)) = (0, g(1))$, $f$ is either zero or piece-wise linear, and so it is piece-wise linear on $[0, g(1)]$.

$\square$

**Proposition 3.6.**    *1. Let $f$ be a nontrivial piece-wise linear tradeoff function with $k \geq 1$ breakpoints and such that $f(x) = 0$ implies that $x = 0$. Then there is no tradeoff function $g$ such that $g^{\circ(k+1)} = f$.*

2. *Let $\epsilon > 0$. There does not exist nontrivial symmetric tradeoff functions $f_1$ and $f_2$ such that $f_{\epsilon,0} = f_1 \circ f_2$.*

3. *Let $f$ be the tradeoff function obtained by an arbitrary sequence of mechanism compositions, functional compositions, or subsampling (without replacement) of $f_{\epsilon,0}$ (could be different $\epsilon$ values for each). Then $f$ is not infinitely divisible and so does not have a log-concave CND.*

*Proof.*    1. By part 3 of Lemma A.12, if $f$ could be written as $g^{\circ(k+1)} = f$, then $g$ must also be piece-wise linear. Since $f(x) = 0$ implies that $x = 0$, $g$ also satisfy $g(x) = 0$ implies $x = 0$. If $g$ is a nontrivial piece-wise linear tradeoff function (with $j \geq 1$ breakpoints), then by part 2 of Lemma A.12, $g^{\circ(k+1)}$ has $j + (k+1) - 1 > k$ break points. This contradicts that $g^{\circ(k+1)} = f$.

2. Suppose that $f_{\epsilon,0} = g \circ h$, where both $g$ and $h$ are non-trivial. By part 3 of Lemma A.12, we know that $h$ is piece-wise linear. Since we are assuming that $h$ is non-trivial, it must have at least one break point. Since $f_{\epsilon,0}(x) = 0$ implies that $x = 0$, $g$ must have this property as well. By Lemma part 1 of A.12, this implies that $h$ must have a single break point. To agree with $f_{\epsilon,0}$, the breakpoint of $h$ must be at $1 - c$, where $c = 1/(1 + \exp(\epsilon))$ is the solution to $f_{\epsilon,0}(1 - c) = c$, since this is where the unique breakpoint of $f_{\epsilon,0}$ lies. However, since $h$ is a symmetric piece-wise linear function with a unique breakpoint at $1 - c$, the only possibility is that $h = f_{\epsilon,0}$.

3. Each application of composition, functional composition, and subsampling without replacement preserves the piece-wise property of the tradeoff function, as well as the property that $f(x) = 0$ implies that $x = 0$. The result follows from part (a).

$\square$

**Proposition 4.2.** *Suppose that $f = f_1 \otimes f_2 \otimes \cdots \otimes f_k$ all be nontrivial and symmetric tradeoff functions, and let $F_1, F_2, \ldots, F_k$ be CNDs for $f_1, \ldots, f_k$ respectively. Let $N = (N_1, \ldots, N_k)$ be the random vector where $N_i \sim F_i$ are independent. Then $N$ is a CND for $f$ with respect to $\|\cdot\|_\infty$.*

*Proof.* For property 1 of Definition 4.1, let $v$ be such that $\|v\|_\infty \leq 1$. Then $v = (v_1, \ldots, v_k)$ is such that $|v_i| \leq 1$. Then

$$
\begin{aligned}
T(N, N + v) &= T((N_1, \ldots, N_k), (N_1 + v_1, \ldots, N_k + v_k)) \\
&= T(N_1, N_1 + v_1) \otimes \cdots \otimes T(N_k, N_k + v_k) \\
&\geq f_1 \otimes \cdots \otimes f_k.
\end{aligned}
$$

If we set $v_i^* = 1$ for all $i$, then repeating the above gives equality in the last step, proving property 2 of Definition 4.1. Call $g(x_1, \ldots, x_n) := F_1'(x_1) \cdots F_k'(x_k)$ the density of $N$. For property 4 of Definition 4.1, since $F_i'$ is symmetric about zero we have that $g$ is also symmetric about zero. For property 3, let $w = (w_1, \ldots, w_k)^\top$ be any vector and $v^* = (1, 1, \ldots, 1)^\top$. Then,

$$\frac{g(w + tv^* - v^*)}{g(w + tv^*)} = \frac{F_1'(w_1 + (t-1)) \cdots F_k'(w_k + (t-1))}{F_1'(w_1 + t) \cdots F_k'(w_k + t)},$$

which is increasing in $t$, since each of the factors is increasing in $t$, by property 3 of Definition 2.2.

$\square$

**Theorem 4.3.** *Let $f$ be a nontrivial and symmetric log-concave tradeoff function with log-concave CND $F$. Let $N = (N_1, \ldots, N_k)$ be the random vector where $N_i \sim F$ are independent. Then $N$ is a (log-concave) CND for $f$ with respect to $\|\cdot\|_1$.*

*Proof.* Since $f$ is a nontrivial log-concave tradeoff function, by Theorem 3.3 there exists a monoid of log-concave tradeoff functions $\{f_t \in \mathscr{F} \mid t \geq 0\}$ satisfying $f_t \circ f_s = f_{t+s}$ such that $f_1 = f$ and $f_t = F(F^{-1}(\alpha) - t)$ for all $t > 0$. Note that for any $t \in \mathbb{R}$, $T(N_i, N_i + t) = f_{|t|}$.

For property 1 of Definition 4.1, let $x$ be such that $\|x\|_1 \leq 1$. Note that $|x_i| < 1$ for all $i = 1, \ldots, k$. Then

$$\begin{aligned}
T(N, N + x) &= f_{|x_1|} \otimes f_{|x_2|} \otimes \cdots \otimes f_{|x_k|} \\
&\geq f_{|x_1|} \circ f_{|x_2|} \circ \cdots \circ f_{|x_k|} \\
&= f_{\sum_{i=1}^k |x_i|} \\
&= f_{\|x_i\|_1} \\
&\geq f_1,
\end{aligned}$$

where in the first line, we use the property that $T(N_i, N_i + x_i) = f_{|x_i|}$ which uses log-concavity, the second line uses Lemma A.2, and the third line uses the property that $f_s \circ f_t$ within the monoid. Note that for $v = (1, 0, 0, \ldots, 0)$, $T(N, N + v) = T(N_1, N_1 + 1) = f$, proving property 2 of Definition 4.1. Since $N$ is constructed by independent 1-d CNDs, the same arguments used in the proof of Proposition 4.2 can be used to prove properties 3 and 4 of Definition 4.1. Note that $N$ is log-concave, since it is a product distribution with log-concave components. $\square$

**Proposition 4.4.** *Let $\Sigma$ be a $d \times d$ positive definite matrix. Let $v^* \in \arg\max_{\|u\| \leq 1} \|\Sigma^{-1/2} u\|_2$. Then $N(0, \Sigma)$ is a $d$-dimensional CND for $\|\Sigma^{-1/2} v^*\|_2$-GDP with respect to the norm $\|\cdot\|$.*

*Proof.* Let $N \sim N(0, \Sigma)$. Note that $\Sigma^{-1/2} N \sim N(0, I_d)$. Let $u$ be such that $\|u\| \leq 1$. Then

$$\begin{aligned}
T(N, N + u) &= T\left(\Sigma^{-1/2} N, \Sigma^{-1/2} N + \Sigma^{-1/2} u\right) \\
&= T\left(N(0, I_d), N(0, I_d) + \Sigma^{-1/2} u\right) \\
&= T\left(N(0, 1), N(\|\Sigma^{-1/2} u\|_2, 1)\right) \\
&= G_{\|\Sigma^{-1/2} u\|_2} \\
&\geq G_{\|\Sigma^{-1/2} v^*\|_2},
\end{aligned}$$

where for the third line, we use the rotational invariance of the multivariate Gaussian distribution. Note that setting $u = v^*$ gives equality. This establishes properties 1 and 2 of Definition 4.1. Property 4 of Definition 4.1 holds since the density of $N(0, \Sigma)$ is symmetric about zero. For property 3, let

$w \in \mathbb{R}^k$ be any vector, and call $g$ the density of $N(0, \Sigma)$. Call $a = (w + tv^*)$ and $b = -v^*$. Then,

$$\log \frac{g(w + tv^* - v^*)}{g(w + tv^*)} = \log \frac{\exp(-\frac{1}{2}(w + (t-1)v^*)^\top \Sigma^{-1}(w + (t-1)v^*))}{\exp(-\frac{1}{2}(w + tv^*)^\top \Sigma^{-1}(w + tv^*))}$$

$$= -\frac{1}{2}\left[ ((w + tv^*) - v^*)^\top \Sigma^{-1}((w + tv^*) - v^*) + (w + tv^*)^\top \Sigma^{-1}(w + tv^*) \right]$$

$$= -\frac{1}{2}\left[ (a + b)^\top \Sigma^{-1}(a + b) - a^\top \Sigma^{-1}a \right]$$

$$= -\frac{1}{2}\left[ a^\top \Sigma^{-1}a + 2a^\top \Sigma^{-1}b + b^\top \Sigma^{-1}b - a^\top \Sigma^{-1}a \right]$$

$$= -\frac{1}{2}\left[ 2a^\top \Sigma^{-1}b + b^\top \Sigma^{-1}b \right]$$

$$= -\frac{1}{2}\left[ 2(w + tv^*)^\top \Sigma^{-1}(-v^*) + (v^*)^\top \Sigma^{-1}v^* \right]$$

$$= t(v^*)^\top \Sigma^{-1}v^* + w^\top \Sigma^{-1}v^* - \frac{1}{2}(v^*)^\top \Sigma^{-1}v^*,$$

which is increasing in $t$, since $\Sigma$ is positive definite, which verifies property 3 of Definition 4.1. $\square$

**Proposition 4.6.** *Let $0 < \delta \leq 1$, $d \geq 1$, and $\|\cdot\|$ be a norm on $\mathbb{R}^d$. Call $v^* \in \underset{\|v\| \leq 1}{\arg \min} \prod_{i=1}^d (1 - \delta|v_i|)$ and $A = \prod_{i=1}^d (1 - \delta|v_i^*|)$. Then $U(\frac{-1}{2\delta}, \frac{1}{2\delta})^n$ is a CND for $f_{0,1-A}$ under $\|\cdot\|$-sensitivity. In the special case of $\|\cdot\| = \|\cdot\|_\infty$, this simplifies to $A = (1 - \delta)^d$.*

*Proof.* Let $X \sim U(\frac{-1}{2\delta}, \frac{1}{2\delta})^n$, and let $v$ be such that $\|v\| \leq 1$. We need a lower bound on $T(X, X+v)$. Since this is the testing of shifted uniforms, $T(X, X+v) = f_{0,\text{TV}(X,X+v)}$.

$$\text{TV}(X, X + v) = 1 - \frac{\prod_{i=1}^d \left( \frac{1}{\delta} - |v_i| \right)}{\delta^{-d}}$$

$$= 1 - \prod_{i=1}^d (1 - \delta|v_i|)$$

$$\geq 1 - A,$$

which establishes property 1 of Definition 4.1. Note that using $v^*$ as defined above, we get that $\text{TV}(X, X + v^*) = 1 - A$, giving property 2 of Definition 4.1. Property 4 of Definition 4.1 is obvious, since each uniform is centered at zero. For property 3, let $w \in \mathbb{R}^d$. The likelihood ratio is

$$\frac{g(w + (t-1)v^*)}{g(w + tv^*)} = \prod_{i=1}^d \frac{I(\frac{-1}{2\delta} \leq w_i + (t-1)v_i^* \leq \frac{1}{2\delta})}{I(\frac{-1}{2\delta} \leq w_i + tv_i^* \leq \frac{1}{2\delta})},$$

and we see that each of these factors can take the possible values:

$$\begin{cases} \text{undefined} & \text{when } w_i + (t-1)v_i^* \notin [\frac{-1}{2\delta}, \frac{1}{2\delta}] \text{ and } w_i + tv_i^* \notin [\frac{-1}{2\delta}, \frac{1}{2\delta}] \\ 0 & \text{when } w_i + (t-1)v_i^* \notin [\frac{-1}{2\delta}, \frac{1}{2\delta}] \text{ and } w_i + tv_i^* \in [\frac{-1}{2\delta}, \frac{1}{2\delta}] \\ 1 & \text{when } w_i + (t-1)v_i^* \in [\frac{-1}{2\delta}, \frac{1}{2\delta}] \text{ and } w_i + tv_i^* \in [\frac{-1}{2\delta}, \frac{1}{2\delta}] \\ +\infty & \text{when } w_i + (t-1)v_i^* \in [\frac{-1}{2\delta}, \frac{1}{2\delta}] \text{ and } w_i + tv_i^* \notin [\frac{-1}{2\delta}, \frac{1}{2\delta}] \end{cases}$$

If $w_i + tv_i^* \in [\frac{-1}{2\delta}, \frac{1}{2\delta}]$ for some $t$, then we have that as $t$ progresses from $-\infty$ to $\infty$, the value of each factor goes from undefined to 0 to 1 to $+\infty$ to undefined, which is a non-decreasing sequence. If $w_i + tv_i^* \notin [\frac{-1}{2\delta}, \frac{1}{2\delta}]$ for every $t$, then the likelihood ratio is always undefined, which is also trivially non-decreasing. We see that property 3 of Definition 4.1 holds. $\square$

The privacy loss random variable is a concept that appears in all major definitions of differential privacy. In fact, Zhu et al. [2022] showed that the privacy loss random variables can be losslessly converted back and forth to the corresponding tradeoff function. For part of the proof of Proposition 4.7, it will be easier to work with the privacy loss random variables than directly with the tradeoff functions. First, we give a formal definition and a few basic properties of privacy loss random variables. While similar results appeared in Zhu et al. [2022], we include them here for completeness.

**Definition A.13** (Privacy Loss Random Variable). Let $X$ and $Y$ be two random variables on $\mathbb{R}^d$, with densities $p$ and $q$, respectively. The *privacy loss random variable* is $\mathrm{PLRV}(X|Y) := \log \frac{p(X)}{q(X)}$, where $X \sim p$.

**Lemma A.14** (Privacy Loss RV is Sufficient). *Let $X \sim p$ and $Y \sim q$ be two random variables on $\mathbb{R}^d$ with densities $p$ and $q$, respectively. Define $L(x) : \mathcal{X} \to \mathbb{R}$ by $L(x) = \log[q(x)/p(x)]$. Note that $L(X) \stackrel{d}{=} -\mathrm{PLRV}(X|Y)$ and $L(Y) \stackrel{d}{=} \mathrm{PLRV}(Y|X)$. Then $T(X,Y) = T(L(X), L(Y)) = T(-\mathrm{PLRV}(X|Y), \mathrm{PLRV}(Y|X))$.*

*Proof.* First, by postprocessing, we have that $T(X,Y) \leq T(L(X), L(Y))$ [Dong et al., 2022, Lemma 2.9]. For the other direction, note that by the Neyman Pearson Lemma, the optimal test for $H_0 : X$ versus $H_1 : Y$ at size $\alpha$ is of the form

$$
\phi(x) = \begin{cases} 1 & L(x) > t \\ c & L(x) = t \\ 0 & L(x) < t, \end{cases}
$$

where $L$ is defined in the Lemma statement, and the values of $t$ and $c$ are uniquely chosen such that $\mathbb{E}_{X \sim p}\phi(X) = \alpha$.

For a specified $t$ and $c$, the type I error is

$$
\begin{aligned}
\text{type I} = \mathbb{E}_{X \sim p}\phi &= \mathbb{E}_{X \sim p}[I(L(X) > t) + cI(L(X) = t)] \\
&= P_{X \sim p}(L(X) > t) + cP_{X \sim p}(L(X) = t),
\end{aligned}
$$

which we see only depends on the statistic $L(X)$. On the other hand,

$$
\begin{aligned}
\text{type II} = 1 - \mathbb{E}_{Y \sim q}\phi(Q) &= 1 - \mathbb{E}_{Y \sim q}[I(L(Y) > t) + cI(L(Y) = t)] \\
&= P_{Y \sim q}(L(Y) \leq t) - cP_{Y \sim q}(L(Y) = t),
\end{aligned}
$$

which we see only depends on the statistic $L(Y)$.

So, when testing $H_0 : L(X)$ versus $H_1 : L(Y)$, using the particular test $\psi(L) = I(L > t) + cI(L = t)$, where the values of $c$ and $t$ are chosen as above, we recover the type I and type II errors of $T(X,Y)$. We conclude that $T(L(X), L(Y)) \leq T(X,Y)$. Combining the inequalities, we have that $T(X,Y) = T(L(X), L(Y))$. The fact that $L(X) = -\mathrm{PLRV}(X|Y)$ and $L(Y) = \mathrm{PLRV}(Y|X)$ follows from the definition of privacy loss random variables. $\square$

**Lemma A.15.** *Let $X \in \mathbb{R}^d$ be a continuous random vector with density $g$, which is symmetric about zero. Then for any $v \in \mathbb{R}^d$, $\mathrm{PLRV}(X|X + v) = \mathrm{PLRV}(X + v|X)$. It follows that*

1. *$T(X, X + v) = T(\mathrm{PLRV(X|X + v)}, -\mathrm{PLRV}(X|X + v))$, and*

2. *Let $Y \in \mathbb{R}^p$ be another continuous random vector symmetric about zero, and let $w \in \mathbb{R}^p$. Then if $\mathrm{PLRV}(X|X + v) \stackrel{d}{=} \mathrm{PLRV}(Y|Y + w)$ then $T(X, X + v) = T(Y, Y + w)$.*

*Proof.* First note that $\mathrm{PLRV}(X|X + v) = \log \frac{g(X)}{g(X-v)}$, where $X \sim g$. Setting $Z = v - X$, we can write

$$
\begin{aligned}
\mathrm{PLRV}(X + v|X) &= \log \frac{g(X - v)}{g(X)}, \text{ where } (X - v) \sim g \\
&= \log \frac{g(-Z)}{g(-Z + v)}, \text{ where } -Z \sim g \\
&= \log \frac{g(Z)}{g(Z - v)}, \text{ where } Z \sim g \quad \text{(by symmetry of } g) \\
&\stackrel{d}{=} \mathrm{PLRV}(X|X + v).
\end{aligned}
$$

Combining the above work with Lemma A.14, we get $T(-\mathrm{PLRV}(X|X + v), \mathrm{PLRV}(X|X + v))$, which is equivalent to statement 1, since the tradeoff function is symmetric. For part two, if

$\mathrm{PLRV}(X|X+v) \overset{d}{=} \mathrm{PLRV}(Y|Y+w)$, then clearly $T(\mathrm{PLRV}(X|X+v), -\mathrm{PLRV}(X|X+v)) = T(\mathrm{PLRV}(Y|Y+w), -\mathrm{PLRV}(Y|Y+w))$, which is equivalent to the statement in part 2, by part 1. $\qquad\square$

**Proposition 4.7.** *Let $\epsilon > 0$, and $d \geq 1$. Let $X$ be a $d$-dimensional random vector with density $g(x) = \frac{\exp(-\epsilon\|x\|_\infty)}{d!(2/\epsilon)^d}$. Then $X$ is a CND for the tradeoff function $L_\epsilon$ with respect to $\|\cdot\|_\infty$.*

*Proof.* Note that for any vector $s \in \{-1, 1\}^d$, $sX \overset{d}{=} X$ (entry-wise multiplication). Because of this, it suffices to consider $T(X, X+v)$ for $v \geq 0$ (all entries non-negative).

First we will show that $T(X, X+1) = T(L, L+1)$, where $X$ is the $\ell_\infty$-mech, and $L \sim \mathrm{Laplace}(0, 1/\epsilon)$ which has density $\frac{\epsilon}{2}\exp(-\epsilon|x|)$. We will do this using privacy loss random variables, applying part 2 of Lemma A.15. Note that since $X$ and $L$ are both symmetric random variables, it suffices to equate the privacy loss random variables $\mathrm{PLRV}(X|X+1)$ and $\mathrm{PLRV}(L|L+1)$. We can easily derive that $\mathrm{PLRV}(L|L+1) = -\epsilon|L| + \epsilon|L-1| = \epsilon[1-2L]_{-1}^1$, where $L \sim \mathrm{Laplace}(0, 1/\epsilon)$ and $[x]_a^b := \min\{\max\{x, a\}, b\}$ is the clamping function. Note that $\mathrm{PLRV}(L|L+1) \overset{d}{=} \epsilon[1 - L_2]_{-1}^1$, where $L_2 \sim \mathrm{Laplace}(2/\epsilon)$.

Now for $T(X, X+1)$, the privacy loss random variable is $\mathrm{PLRV}(X|X+1) = -\epsilon\|X\|_\infty + \epsilon\|X-1\|_\infty$, where $X \sim g(x)$. We can simplify this expression as follows, using the notation $\max(X) = \max_{1 \leq i \leq d} X_i$ and $\min(X) = \min_{1 \leq i \leq d} X_i$:

$$\mathrm{PLRV}(X|X+1)$$
$$= -\epsilon\|X\|_\infty + \epsilon\|X-1\|_\infty$$
$$= \begin{cases} -\epsilon\max(X) + \epsilon(1-\min(X)) & \text{if } \max(X) \geq -\min(X) \text{ \& } 1-\min(X) \geq \max(X)-1 \\ -\epsilon\max(X) + \epsilon(\max(X)-1) & \text{if } \max(X)-1 > 1-\min(X) \\ -\epsilon(-\min(X)) + \epsilon(1-\min(X)) & \text{if } -\min(X) > \max(X) \end{cases}$$
$$= \begin{cases} -\epsilon\max(X) + \epsilon(1-\min(X)) & \text{if } -1 \leq [1-(\max(X)+\min(X))] \leq 1 \\ -\epsilon & \text{if } [1-(\max(X)+\min(X))] < -1 \\ \epsilon & \text{if } [1-(\max(X)+\min(X))] > 1 \end{cases}$$
$$= \epsilon[1-(\max(X)+\min(X))]_{-1}^1.$$

Comparing this expression with $\mathrm{PLRV}(L|L+1)$, we see that it suffices to show $\max(X)+\min(X) \overset{d}{=} \mathrm{Laplace}(2/\epsilon)$. Recall that $X \overset{d}{=} RU$, where $R \sim \mathrm{Gamma}(d+1, \epsilon)$, using the shape, rate parameterization, and $U_i \overset{iid}{\sim} U(-1, 1)$ for $i = 1, \ldots, d$ [Hardt and Talwar, 2010, Remark 4.2]. By factoring out $R$, we get

$$\max(X) + \min(X) \overset{d}{=} R(\max(U) + \min(U)).$$

So, we will determine the distribution of $\max(U) + \min(U)$ first. We can easily compute the joint distribution of $\max(U)$ and $\min(U)$, as these are the minimum and maximum order statistics:

$$f_{\min(U),\max(U)}(x, y) = \frac{d(d-1)}{4}\left(\frac{y-x}{2}\right)^{d-2} I(-1 \leq x \leq y \leq 1).$$

Now consider the change of variables $m = x$ and $w = x + y$. Applying change of variables, we have

$$f_{\min(U),\max(U)+\min(U)}(m, w) = \frac{d(d-1)}{4}\left(\frac{(w-m)-m}{2}\right)^{d-2} I(-1 \leq m \leq w - m \leq 1)$$
$$= \frac{d(d-1)}{2^d}(w-2m)^{d-2} I(-1 \leq m, m \leq w/2, w - 1 \leq m).$$

To get the distribution of $W = \max(U) + \min(U)$, we marginalize out $M = \min(U)$:

$$f_{\max(U)+\min(U)}(w) = \int_{\max\{-1, w-1\}}^{w/2} \frac{d(d-1)}{2^d} (w - 2m)^{d-2} \, dm$$

$$= \frac{d(d-1)}{2^{d+1}} \left. \frac{-(w-2m)^{d-1}}{d-1} \right|_{\max\{-1,w-1\}}^{w/2}$$

$$= \frac{d}{2^{d+1}} \left[ (w - 2\max\{-1, w-1\})^{d-1} - \cancel{(w - 2(w/2))^{d-1}} \right]$$

$$= \frac{d}{2^{d+1}} \begin{cases} (w+2)^{d-1} & -2 \leq w \leq 0 \\ (2-w)^{d-1} & 0 \leq w \leq 2 \end{cases}$$

$$= \frac{d}{2^{d+1}} (2 - |w|)^{d-1} I(-2 \leq w \leq 2).$$

Since the distribution of $W = \max(U) + \min(U)$ is symmetric about zero, $W \stackrel{d}{=} (-1)^B |W|$ where $B \sim \text{Bern}(1/2)$. So, our goal is to show $(-1)^B RW \stackrel{d}{=} (-1)^B \text{Exp}(\epsilon/2)$, since the left side is equal in distribution to $\max(X) + \min(X)$ and the right side is equal in distribution to $\text{Laplace}(2/\epsilon)$. The pdf of $Y :\stackrel{d}{=} |W|$ is $f(y) = \frac{d}{2^d}(2-y)^{d-1} I(0 \leq y \leq 2)$. It suffices to show that $RY \stackrel{d}{=} \text{Exp}(\epsilon/2)$. Let $\phi_R$ be the characteristic function of $R$ and $\phi_{RY}$ be the characteristic function of $RY$. Then,

$$\phi_{RY}(t) = \mathbb{E}_Y \phi_R(tY)$$

$$= \mathbb{E}_Y \left( 1 - \frac{ity}{\epsilon} \right)^{-(d+1)}$$

$$= \frac{d}{2^d} \int_0^2 \left( 1 - \frac{ity}{\epsilon} \right)^{-(d+1)} (2-y)^{d-1} \, dy$$

$$= \frac{\epsilon}{\epsilon - 2it}$$

$$= \frac{\epsilon/2}{\epsilon/2 - it},$$

which we identify as the characteristic function of $\text{Exp}(\epsilon/2)$, establishing that $RY \stackrel{d}{=} \text{Exp}(\epsilon/2)$. By part 2 of Lemma A.15, this completes the proof that $T(X, X+1) = T(L, L+1)$, establishing property 2 of Definition 4.1. Note that property 4 of Definition 4.1 is obvious, and property 3 holds since the likelihood ratio $g(x-1)/g(x)$ is an increasing function in $\max(x) + \min(x)$, which itself is an increasing function of $t$ when $x = w + t$ for every vector $w \in \mathbb{R}^d$. It remains to verify property 1 of Definition 4.1.

Next we will show that for $v \in (0, 1]^d$, $T(X, X+1) \leq T(X, X+v)$ (this proof strategy is based on the proof of Lemma 3.5 in Dong et al. [2021]). We will separately address the cases that some of $v_i = 0$ at the end of the proof. Call $f_1 = T(X, X+1)$ and $f_v = T(X, X+v) = T(X/v, X/v + 1)$. Define the two linear maps $r : x \mapsto x/v$ and $r^{-1} : x \mapsto vx$ (entry-wise multiplication and division), which are inverse maps. Note that $r(X) = X/v$ has density proportional to $\exp(-\epsilon\|vt\|_\infty)$. Let $\alpha \in [0, 1]$ be given. Let $A$ be the optimal rejection region for $T(X/v, X/v + 1)$ at type I error $\alpha$. By our earlier work, we know that

$$A = \{x \mid \epsilon(\max(vx) + \min(vx)) \geq t\},$$

for some $t \in \mathbb{R}$, and it satisfies $P(r(X) \in A) = \alpha$ and $P(r(X) + 1 \notin A) = f_v(1 - \alpha)$. We can now consider $r^{-1}(A)$ as a possible rejection region for testing $T(X, X+1)$, which is at best suboptimal. We compute the type I error as

$$P(X \in r^{-1}(A_v)) = P(r(X) \in A) = \alpha.$$

Suboptimality of the rejection region implies that

$$
\begin{aligned}
f_1(1-\alpha) &\le P(X+1 \notin r^{-1}(A)) \\
&= P(r(X) + r(1) \notin A) \\
&= P(r(X) + 1/v \notin A) \\
&= P(r(X) \notin A - 1/v) \\
&\le P(r(X) \notin A - 1) \\
&= f_v(1-\alpha),
\end{aligned}
$$

where we used the fact that $r$ is a linear map, and $r(1) = 1/v$; the key step is the final inequality, which we justify as follows: it suffices to show that $(A - 1/v)^c \subset (A - 1)^c$ or equivalently $A - (1/v - 1) \supset A$. We verify this by inspecting the definition of A:

$$
\begin{aligned}
A &= \{x \mid \max(vx) + \min(vx) \ge t\} \\
&\subset \{x \mid \max(vx + (1/v - 1)) + \min(vx + (1/v - 1)) \ge t\} \\
&= \{x - (1/v - 1) \mid \max(vx) + \min(vx) \ge t\} \\
&= A - (1/v - 1),
\end{aligned}
$$

where in the inclusion step, we used the fact that $(1/v - 1) \ge 0$ implies that $\max(vx + (1/v - 1)) \ge \max(vx)$ and $\min(vx + (1/v - 1)) \ge \min(vx)$. This completes the argument that for $v \in (0,1]^d$, $f_1 = T(X, X+1) \le T(X, X+v) = f_v$.

Finally, let $v \in [0,1]^d$, where the entries may possibly be zero. Let $v_n \in (0,1]^d$ be a sequence of vectors converging to $v$. Notice that $X + v_n \overset{TV}{\to} X + v$, since $X$ has a continuous density. Since $T(X, X + v_n) \ge T(X, X+1)$ by our above work, by Corollary A.9 we have $T(X, X+v) \ge T(X, X+1)$ as well. $\qquad\square$

**Theorem 4.8.** *Let $d \ge 2$ and let $\|\cdot\|$ be any norm on $\mathbb{R}^d$. Then for any $\epsilon > 0$, there is no random vector satisfying properties 1 and 2 of Definition 4.1 for $f_{\epsilon,0}$ with respect to the norm $\|\cdot\|$. In particular, there is no multivariate CND for $f_{\epsilon,0}$.*

The proof strategy of Theorem 4.8 is as follows: 1) observe that property of Definition 4.1 enforces constraints on the likelihood ratio $\log \frac{g(x-v)}{g(x)}$, 2) establish that the measure induced by $g$ is equivalent to Lebesgue measure, which simplifies some measure theory details, 3) show that we can construct a vector $w$ such that $\|w\| < 1$, $\|w + v\| < 1$, and $w \notin \mathrm{Span}(v)$, 4) based on the properties of $w$ and $v$, show that by taking integral combinations of $w$ and $v$, we can find an arbitrarily long sequence of points each sufficiently far from each other such that the value of $g$ is bounded below by a common constant, and 5) show that point 4 implies that $g$ is not integrable. Because densities are only well defined up to sets of Lebesgue measure zero, the details of the proof are more complicated to ensure that we are careful about the measure theoretical details.

*Proof.* Suppose to the contrary that there exists a CND for $f_{\epsilon,0}$ with respect to $\|\cdot\|$, which has density $g$. We will denote $\mu_g$ as the measure induced by $g$: $\mu_g(S) = \int_S g(x)\,dx$, and use $\lambda$ to denote Lebesgue measure.

By property 2 of Definition 4.1, there exists $v \in \mathbb{R}^d$ such that $\|v\| \le 1$ and $T(N, v+N) = f_{\epsilon,0}$, where $N \sim g$. This implies that $\log \frac{g(x-v)}{g(x)} = \pm\epsilon$ almost everywhere ($\mu_g$) for all $x \in \mathbb{R}^d$ (if $P$ and $Q$ are two distributions satisfying $T(P, Q) = f_{\epsilon,0}$, then the privacy loss random variable is a binary random variable, taking values in $\{-\epsilon, \epsilon\}$). Furthermore, by property 1, for any other vector $w \in \mathbb{R}^d$ such that $\|w\| \le 1$, we have that $\log \frac{g(x-w)}{g(x)} \in [-\epsilon, \epsilon]$ for almost every $x \in \mathbb{R}^d$ ($\mu_g$).

Before we begin our main argument, we will show that (if such a $g$ exists,) $\mu_g$ must be equivalent to Lebesgue measure. We know that Lebesgue measure dominates $\mu_g$, so we only need to show that $\mu_g = 0$ implies $\lambda = 0$. Suppose to the contrary that there exists $S \subset \mathbb{R}^d$ such that $\lambda(S) > 0$ but $\mu_g(S) = 0$ (which implies that $g(x) = 0$ a.e. on $S$). We claim that there exists such an $S$ such that for some $\|t\| \le 1$, $\mu_g(S + t) > 0$. We prove this as follows: begin with any $S$ such that $\lambda(S) > 0$ but $\mu_g(S) = 0$. If $\mu_g(S + t) = 0$ for all $\|t\| \le 1$, then set $S' = \bigcup_{\|t\| \le 1}(S + t)$, which is strictly larger than $S$. If $S'$ still does not have the desired property, repeat the process iteratively. Note that

in the limit, this process results in $\mathbb{R}^d$, but $\mu_g(\mathbb{R}^d) = 1$. So, the process must terminate, giving us the desired set with the properties $\lambda(S) > 0$, $\mu_g(S) = 0$ and there exists some $\|t\| \leq 1$ such that $\mu_g(S + t) > 0$. Then there exists $P \subset S + t$ such that $g(x) > 0$ on $P$ a.e., and note that $P - t \subset S$ and $g(x) = 0$ on $P - t$ a.e.. However, this implies that $\log \frac{g(x-t)}{g(x)} = \infty \notin [-\epsilon, \epsilon]$ on the set $P$, which has positive probability $\mu_g(P) > 0$. This contradicts property 1 of Definition 4.1, as discussed above. We conclude that $\mu_g$ and $\lambda$ are equivalent measures. So, we will interchangeably use statements about Lebesgue measure and $\mu_g$ measure.

Let $r > 0$ be the largest value such that $\|x\|_2 \leq r$ implies that $\|x\| \leq 1$ (possible by the equivalence of norms on $\mathbb{R}^d$). Consider three sets

$$A = \{w \mid \|\mathrm{Proj}_{v^\perp} w\|_2 < r/4 \ \& \ \|w\| < 1\},$$
$$B = \{w \mid \|\mathrm{Proj}_{v^\perp} w\|_2 \leq r/8 \ \& \ \|w\| < 1\},$$
$$C = \{w \mid \|w + v\| < 1\}.$$

Note that $(A \setminus B) \cap C$ is an open set; we will demonstrate that it is non-empty, which implies that it has non-zero Lebesgue measure. First note that $A \setminus B \neq \emptyset$, since $d \geq 2$ implies that $\emptyset \subsetneq B \subsetneq A$. We will construct a vector $w \in (A \setminus B) \cap C$ as follows: let $y \in A \setminus B$, and call $z = \mathrm{Proj}_{v^\perp} y$. Then $r/8 < \|z\|_2 < r/4$. We set $w = z - v/2$. First we will check that $w \in A \setminus B$: since $z = \mathrm{Proj}_{v^\perp} w = \mathrm{Proj}_{v^\perp} y$, we have that $r/8 < \|\mathrm{Proj}_{v^\perp} w\| < r/4$. We also need to check that

$$\|w\| = \|z - v/2\| \leq \|z\| + \frac{1}{2}\|v\| \leq \frac{1}{4} + \frac{1}{2} < 1,$$

since $\|z\|_2 \leq r/4$ implies that $\|z\| \leq 1/4$. Next, we will check that $w \in C$:

$$\|w + v\| = \|z - v/2 + v\| = \|z + v/2\| \leq \|z\| + \frac{1}{2}\|v\| \leq \frac{1}{4} + \frac{1}{2} < 1,$$

using again the fact that $\|z\|_2 \leq r/4$ implies that $\|z\| \leq 1/4$. We conclude that $(A \setminus B) \cap C$ is a non-empty open set, which implies that it has non-zero Lebesgue measure.

Let $c := \mathrm{esssup}\, g$. Note that $c < \infty$, as otherwise, this would violate the log-likelihood ratio property discussed earlier. Then $g(x) \leq c$ holds with probability one. So, since $(A \setminus B) \cap C$ has positive probability, we can find a vector $w \in (A \setminus B) \cap C$ which satisfies $g(w) \leq c$.

Let $\eta \in (0, c)$ be given. Then the set $\{x \mid c - \eta \leq g(x) \leq c\}$ has positive measure. For $v$ as defined above, and an arbitrary vector $u \in \mathbb{R}^d$, consider two more sets:

$$G_v = \left\{x \,\middle|\, \log \frac{g(x+v)}{g(x)} \in \{-\epsilon, \epsilon\}\right\},$$
$$F_u = \left\{u \,\middle|\, \log \frac{g(x+u)}{g(x)} \in [-\epsilon, \epsilon]\right\},$$

which both hold with probability one whenever $\|u\| \leq 1$.

Let $K$ be a positive integer such that $K > \left(e^{-\epsilon}(c - \eta)\frac{\pi^{d/2}}{\Gamma(1+d/2)}(r/8)^d\right)$. Then there exists $\xi \in \mathbb{R}^d$ such that for every $0 \leq j \leq k \leq K$,

$$\xi \in G_v - (kw + jv),$$
$$\xi \in F_w - (kw + jv),$$
$$\xi \in F_{w+v} - (kw + jv),$$
$$c - \eta \leq g(\xi) \leq c,$$
$$\|\xi\|_2 \leq b := \mathrm{esssup}_{c-\eta \leq g(x) \leq c} \|x\|_2,$$

since the first three lines hold with probability one, the fourth holds with positive probability as discussed earlier, and the last holds with probability one. Note that $b < \infty$ as otherwise, we would have an unbounded region with positive probability such that $g \geq c - \eta > 0$, which would imply that $g$ is not integrable.

Since $\xi \in F_w$, we have that $g(\xi + w) \in [e^{-\epsilon}g(\xi), c] \subset [e^{-\epsilon}(c - \eta), c]$, since $\|w\| \leq 1$. Similarly, as $\xi \in F_{w+v}$, we have that $g(\xi + w + v) \in [e^{-\epsilon}(c - \eta), c]$, since $\|w + v\| \leq 1$. However, since

$\xi \in G_v - w$, we have that $\frac{g(\xi+w+v)}{g(\xi+w)} = e^{\pm\epsilon}$. The only possibility to satisfy all of these constraints is for either $g(\xi + w) \geq c - \eta$ or for $g(\xi + w + v) \geq c - \eta$. Repeating the previous argument, starting with $g(\xi + w) \geq c - \eta$ gives either $g(\xi + 2w) \geq c - \eta$ or $g(\xi + 2w + v) \geq c - \eta$. If instead, we start with $g(\xi + w + v) \geq c - \eta$, then either $g(\xi + 2w + v) \geq c - \eta$ or $g(\xi + 2w + 2v) \geq c - \eta$. We see that after $k$ steps of this procedure, we have that $g(\xi + kw + jv) \geq c - \eta$ for some $j \in \{0, 1, 2, \ldots, k\}$. We denote by $j(k)$ the value of $j$ obtained by this procedure at the $k^{th}$ step.

For each $0 \leq k \leq K$, define

$$A_k = \{x \mid \|x - (\xi + kw + j(k)v)\|_2 < r/8\}.$$

Note that since $w \in A \setminus B$ from above, we know that $\|\text{Proj}_{v^\perp} w\|_2 \geq r/8$. This implies that each $A_k$ is disjoint from the others, since the $A_k$ are of radius $r/8$, and the distance between each set is at least $r/8$. Furthermore, notice that on each $A_k$, $g \geq e^{-\epsilon}(c - \eta)$.

Finally, consider the integral of $g$, which we lower bound:

$$
\begin{aligned}
\int_{\mathbb{R}^d} g(x)\, dx &\geq \sum_{k=0}^{K} \int_{A_k} g(x)\, dx \\
&\geq \sum_{k=0}^{K} \int_{A_k} e^{-\epsilon}(c - \eta)\, dx \\
&= \sum_{k=0}^{K} e^{-\epsilon}(c - \eta)\text{Vol}(A_k) \\
&= (K + 1)e^{-\epsilon}(c - \eta)\frac{\pi^{d/2}}{\Gamma(1 + d/2)}(r/8)^d \\
&> 1,
\end{aligned}
$$

where we used the formula for a $d$-dimensional sphere of radius $r/8$ to evaluate $\text{Vol}(A_k)$, and in the last line, we used the fact that $K > \left(e^{-\epsilon}(c - \eta)\frac{\pi^{d/2}}{\Gamma(1+d/2)}(r/8)^d\right)$. We see that $g$ cannot integrate to one, which contradicts our assumption that it is a multivariate CND. In fact, $g$ is not even integrable, as $K$ could have been chosen arbitrarily high.

$\square$

**Corollary 4.9.** *Let $\epsilon > 0$ be given. There does not exist nontrivial symmetric tradeoff functions $f_1$ and $f_2$ such that $f_{\epsilon,0} = f_1 \otimes f_2$.*

*Proof.* Suppose to the contrary that there did exist a nontrivial decomposition $f_{\epsilon,0} = f_1 \otimes f_2$. Then Proposition 4.2 gives a construction for a 2-dimensional CND of $f_{\epsilon,0}$. However, we know from Proposition 4.8 that there is no 2-dimensional CND for $f_{\epsilon,0}$. $\square$