# OpenReview forum: "Log-Concave and Multivariate Canonical Noise Distributions for Differential Privacy"
_NeurIPS.cc/2022/Conference — NeurIPS 2022 Accept_

### Official Review · Reviewer_RBrs · 2022-07-10

**Rating:** 7
**Confidence:** 3
**Soundness:** 4 excellent
**Presentation:** 4 excellent
**Contribution:** 3 good

**Summary:**

This work investigates canonical noise distributions (CND) of $f$-DP, a generalization of $(\epsilon, \delta) $-DP which characterizes the level of privacy by the hardness of hypothesis tests. CND is a family of distributions that satisfies $f$-DP with no loss in privacy budget. The authors prove the existence and infinite divisibility of log-concave CND and extend the notion of CND to high dimensions. Relations to other notions of DP, such as $(\epsilon, \delta) $-DP, Gaussian DP are also discussed.

**Questions:**

Minor questions
- What is $G_1$ in line 227?


**Limitations:**

Limitations and negative impact are adequately addressed.

**Strengths And Weaknesses:**

Strengths
- Understanding the optimal noise adding strategy is a fundamental theoretical problem for differential privacy. CND satisfies
- This work extends previous work by showing the existence of log-concave CND in 1 dimension and give an equivalent condition (infinite divisibility)
- The authors provide a generalization of CND in higher dimensions and shows that CND can be easily constructed when the tradeoff function can be decomposed into $k$ components.
- Under the framework of $f$-DP, CNDs for various existing privacy notions are derived. The results draw connection to many existing DP mechanisms.
- The writing is clear and well structured.

Weakness
- For $(\epsilon, \delta)$-DP with $\delta > 0$, $\ell_2$ sensitivity is also used frequently. Can we say something about CND for $(\epsilon, \delta)$-DP with respect to $\ell_2$ sensitivity?

---

> ### Author Response · Authors · 2022-07-31
> **Response to Reviewer RBrs**
>
> Reviewer RBrs asks whether there exists a multivariate CND for $(\epsilon,\delta)$-DP with respect to $\ell_2$ sensitivity. We agree that this is an interesting question, but do not currently have any results in this direction. That said, due to the impossibility result for pure-DP, we suspect that there are limited multivariate CNDs for $(\epsilon,\delta)$-DP. In general, an interesting direction for future work is to determine when multivariate CNDs exist for both a given tradeoff function and a given sensitivity norm.
>
> RBrs asks what $G_1$ means in line 227. $G_1$ represents the tradeoff function $T(N(0,1), N(1,1))$, which is the tradeoff function for 1-GDP. This is defined in line 119 of Section 2. In the revision, we will remind the reader that $G_1$ represents this tradeoff function.

---

> > ### Comment · Reviewer_RBrs · 2022-08-09
> > **Thanks for the response.**
> >
> > Thanks very much for your response. I'm keeping my score.

---

### Official Review · Reviewer_yTW3 · 2022-07-11

**Rating:** 6
**Confidence:** 3
**Soundness:** 4 excellent
**Presentation:** 4 excellent
**Contribution:** 3 good

**Summary:**

The paper considers f-DP and canonical noise distributions. f-DP is based on the concept of tradeoff functions which give an order for the distinguishability of pairs of random variables (Dong et al.) and is in a sense stronger notion than (eps,delta)-DP (is in a sense equivalent to privacy loss random variables and so called pairs of dominating distributions, however). Canonical noise distributions 'fill' a given f-DP profile by giving a noise adding mechanism which has exactly the f-DP distinguishability for pairs of additive-noise outputs (with given distance) (canonical noise distributions: Awan and Vadhan). The distributions of Awan and Vadhan are 'ugly', they can be e.g. discontinuous. Dong et al. show that 'nice' mechanisms are those where the noise has a log concave distribution. It is a natural question to ask, which f-DP profiles exhibit log concave canonical distributions and how do canonical noise distributions look for multivariate distributions.

One of the weaknesses of DP is that it is hard to interpret and also a very hard topic for non-experts. GDP (and f-DP in general) is good in a sense that it is easy to understand (which one is more distinguishable) and easy to calculate with (simply add up the mu's). While accurate DP accountants and analyses are important, it is important to make DP as easily understand ( on a rigorous grounding) as possible. So in this perspective this line of work is really important.

**Questions:**

I find the subsampling part of the following statement a bit unexpected (Proposition 3.6) :
"Let f be the tradeoff function obtained by an arbitrary sequence of mechanism compositions, functional compositions, or subsampling (without replacement) of $f_{\varepsilon,0}$ (could be different values for each). Then f is not infinitely divisible and so does not have a log-concave CND."

Why is that statement included, what is its value?

On a related note, I was hoping to see some statements about subsampling. Can you comment: is there a way to characterise the subsampled versions of log concave trade off functions? The composition results would be then some CLT - based results as in case of GDP, right?


Comment: Typo, line 169: "In Section 3.1 that the Tulap distribution is the unique CND...."

**Limitations:**

From the applicability perspective the work is quite limited: as far as I see only the GDP related results are directly practical. However this is basic research so I don't see that as a big deficit.

**Strengths And Weaknesses:**

Strengths:

- Strong theory paper which continues this important line of work on f-DP. The math seems all correct and the writing is on a high level.

- Interesting connection between group privacy and compositions that has to hold (similar property holds particularly for GDP) for a tradeoff function f to have log concave canonical distributions.

Weaknesses:

- The most practical looking results concern Gaussian DP and their novelty does not seem that big (e.g. covariance matrix Sigma in multivariate Gaussian noise corresponds to scaling the sensitivity appropriately). So I am not entirely sure whether the characterisations given by the paper will have that much practical value.

- Still I think quite limited contribution. This is a strong theory paper and DP community needs this ground work

---

> ### Author Response · Authors · 2022-07-31
> **Response to Reviewer yTW3**
>
> Reviewer yTW3 states that "The most practical looking results concern Gaussian DP and their novelty does not seem that big". While the result on the Gaussian mechanism may not be very surprising, we claim that our main contribution is to the basic research on CNDs, fundamental properties of tradeoff functions, and the connection between group privacy, composition, and the construction of CNDs. While the framework of GDP has many desirable properties, there are still many reasons why one may be interested in other DP frameworks. In some applications, having a stronger notion of DP is needed to protect events with small probability, such as pure-DP or Laplace-DP; in this case, our work shows that Laplace-DP is a much better behaved notion of privacy than pure-DP. One may also propose other alternative DP definitions based on a family of tradeoff functions, and our research gives some fundamental insights on what properties that family must have in order for multivariate (and log-concave) CNDs to be constructed.
>
> Reviewer yTW3 asks about the statement of Proposition 3.6. We clarify that any of the operations 1) mechanism composition, 2) functional composition, and 3) subsampling (without replacement) preserve both the piece-wise linear property and the property $f(x)=0$ implies $x=0$; as such, statement 1 of Proposition 3.6 implies that the resulting tradeoff function is not infinitely divisible. The purpose of this statement is to emphasize that not only does pure DP lack a log-concave CND, but so do several related tradeoff functions, highlighting the limitations of pure DP.
>
> Reviewer yTW3 also asks about subsampling applied to log-concave tradeoff functions. It is easy to verify that a piece-wise linear tradeoff function (with $f(x)=0$ implies $x=0$) is preserved under subsampling (without replacement). However, beyond this, it is not clear whether infinite-divisibility is preserved under subsampling or not. The reviewer may be interested in reading Section 4 of "Gaussian Differential Privacy" (https://arxiv.org/pdf/1905.02383.pdf), which gives a general formula for subsampling applied to a tradeoff function; in section 5, they also apply the CLT to do composition on the subsampled tradeoff functions. It does not seem that they used the log-concavity assumption in their derivation, so it is not clear whether log-concavity has any desirable properties related to subsampling.

---

> > ### Comment · Reviewer_yTW3 · 2022-08-09
> > **Thank you for the response**
> >
> > Thank you for addressing all my questions, I am keeping my score (voting for acceptance).

---

### Official Review · Reviewer_B1pt · 2022-07-12

**Rating:** 7
**Confidence:** 4
**Soundness:** 4 excellent
**Presentation:** 3 good
**Contribution:** 4 excellent

**Summary:**

This paper studies distributions that can be used as additive noise mechanisms, which tightly/exactly satisfy f-differential privacy but also enjoy nice properties such as being log-concave or multivariate. Such tight distributions are called canonical noise distributions (CDNs) for f-DP. First, the paper gives a sufficient and necessary set of conditions on f for the existence of log-concave CNDs for f-DP and on the one hand proves that a CND for any piece-wise linear f as well as the unique CND for f_(epsilon,0) (pure DP) is not log-concave and on the other hand observes that f_(0,delta) is log-concave. Secondly, the paper extends the definition to multivariate CNDs and gives examples of CNDs for the special cases of f that corresponds to pure-, approximate-, (0,delta)-, Gaussian-, and Laplace-DP for different sensitivity norms (ell-1,2,infinity).

**Questions:**

In general I liked the paper and I think it has solid contributions.

One suggestion I have for the presentation is to present the contributions more clearly in the introduction of the main paper, instead of mentioning them all in parallel with the organization. I think it would be clearer to break them down into the log-concave and multivariate results, and make some of the intuition about group privacy/composition, which is mentioned more vaguely before that, more clear.

Small typos:
abstract: Gaussian-DP. -> Gaussian-DP,
line 140: |S(D)-S(D') | -> ||S(D)-S(D') ||

**Limitations:**

No apparent potential negative societal impact. Some technical limitations have been sufficiently discussed.

**Strengths And Weaknesses:**

Strengths:
- f-DP is a general definition that can be translated to several DP definitions that see extended use. Given this generality, I think that the pursuit of optimal additive noise distributions (CNDs) (in the sense that they satisfy exactly the f-DP definition for some worst-case set of datasets) is useful and interesting.
- This paper gives concrete extensions over the first paper in this area, which defined CNDs and focused on univariate noise distributions. The extensions are two-fold. First, in studying for which types of f we can have log-concave CNDs for f-DP and providing examples and an impossibility result for pure DP. Second, in extending the definition to multivariate noise distributions. Both (log-concave and multivariate) are desirable properties of noise distributions used for DP, and especially the latter seems like an interesting direction for future work.

Weaknesses:
- The results on log-concave distributions seem more complete/conclusive but this is not the case for multivariate CNDs, which I think is the more interesting contribution, especially for DP high-dimensional statistics. However, I do think the examples provided are a good start and given that this paper is the first to extend the definition to the multivariate case and the log-concave conditions are fully characterized, this is a solid contribution as is.
- The paper builds directly on recent work and in a somewhat niche area. I think this along with the space constraints make for a quite dense paper that seems targeted to a rather small audience. I liked some choices in terms of the presentation (I appreciated the proof sketches and illustration) but in general it is easy to loose the big picture in the text. I think this is a minor issue.

---

> ### Author Response · Authors · 2022-07-31
> **Response to Reviewer B1pt**
>
> We agree with Reviewer B1pt that the results for multivariate CNDs are only the beginning of this area of research. We believe that our most interesting contributions in this direction are the collection of positive and negative results for even the existence of multivariate CNDs. We found it surprising that pure DP does not have any multivariate CND, whereas all of the other considered notions of DP do have multivariate CNDs. We believe that it is an interesting direction of future work to determine when multivariate CNDs exist for specific sensitivity norms and tradeoff functions, as well as to explore the variety of different multivariate CNDs available.
>
> Reviewer B1pt also points out that the research is in a niche area and that the big picture may be lost to the reader. While we agree that the area of $f$-DP/CNDs is a new area of research, we believe that this is an important and fundamental area of privacy research. In "Gaussian Differential Privacy", they argued that $f$-DP is the most general privacy framework, in that any privacy notion which satisfies the data processing inequality can be derived from a tradeoff function. We believe that the development of CNDs (which optimally satisfy the privacy notion) is an important area of research that will have a significant impact in the field. As reviewer B1pt suggests, we will modify the introduction to make our contributions clearer, and explain the intuition behind group privacy and composition.
>
> We thank Reviewer B1pt for catching the sensitivity typo. This will be corrected.

---

### Author Response · Authors · 2022-07-31
**Thank you & highlight of contributions**

We are thankful for the reviewers' comments and suggestions, and are glad that all three reviewers view the problems of studying log-concave and multivariate CNDs as important and interesting. We hope that the discussion in this rebuttal period will clarify our contributions.

Before responding to each reviewer, we want to highlight the contributions of this paper: This is the first work to develop log-concave or multivariate distributions which exactly match the privacy guarantee, and we have developed concrete positive and negative results in both areas. We showed that a log-concave CND exists if and only if the tradeoff function is infinitely-divisible (and we give a construction), a property that we introduced which is related to group privacy. We also showed that certain piece-wise linear tradeoff functions (including pure-DP) are not log-concave. We proposed a definition for multivariate CNDs, which extends the definition of Awan & Vadhan, and showed that if a tradeoff function is either infinitely-divisible, or decomposable, then multivariate CNDs exist (and we give constructions). We also showed that there is no multivariate CND for pure-DP under any sensitivity norm. These results give insight into the structure of $f$-DP, and how group privacy and mechanism composition are related to the existence and construction of optimal noise mechanisms. While this work is focused on theory, we believe that the results of this paper give a deeper understanding of $f$-DP, and that the constructions of optimal noise mechanisms will contribute to greater advances in DP research.

---

### Meta-Review · Area_Chair_X3mb · 2022-08-27

**Recommendation:** Accept
**Confidence:** Certain

**Metareview:**

This paper provides conditions for the existence of log-concave multivariate distributions to satisfy f-differential privacy constraints. The results of the paper have the potential to be broadly applicable.

**Award:**

No

---

### Decision · Program_Chairs · 2022-09-14

Accept